# Diffusion-based Adversarial Purification
# from the Perspective of the Frequency Domain

**Gaozheng Pei** [1]  **Ke Ma** [1]  **Yingfei Sun** [1]  **Qianqian Xu** [2]  **Qingming Huang** [3][2][4]

## Abstract

The diffusion-based adversarial purification methods attempt to drown adversarial perturbations into a part of isotropic noise through the forward process, and then recover the clean images through the reverse process. Due to the lack of distribution information about adversarial perturbations in the pixel domain, it is often unavoidable to damage normal semantics. We turn to the frequency domain perspective, decomposing the image into amplitude spectrum and phase spectrum. We find that for both spectra, the damage caused by adversarial perturbations tends to increase monotonically with frequency. This means that we can extract the content and structural information of the original clean sample from the frequency components that are less damaged. Meanwhile, theoretical analysis indicates that existing purification methods indiscriminately damage all frequency components, leading to excessive damage to the image. Therefore, we propose a purification method that can eliminate adversarial perturbations while maximizing the preservation of the content and structure of the original image. Specifically, at each time step during the reverse process, for the amplitude spectrum, we replace the low-frequency components of the estimated image's amplitude spectrum with the corresponding parts of the adversarial image. For the phase spectrum, we project the phase of the estimated image into a designated range of the adversarial image's phase spectrum, focusing on the low

[1]School of Electronic, Electrical and Communication Engineering, UCAS, Beijing. [2]Key Laboratory of Intelligent Information Processing, Institute of Computing Technology, CAS, Beijing. [3]School of Computer Science and Technology, UCAS, Beijing. [4]Key Laboratory of Big Data Mining and Knowledge Management, UCAS, Beijing.. Correspondence to: Ke Ma <make@ucas.ac.cn>, Qingming Huang <qmhuang@ucas.ac.cn>.

*Proceedings of the 42nd International Conference on Machine Learning*, Vancouver, Canada. PMLR 267, 2025. Copyright 2025 by the author(s).

frequencies. Empirical evidence from extensive experiments demonstrates that our method significantly outperforms most current defense methods. Code is available at https://github.com/GaozhengPei/FreqPure.

## 1. Introduction

Adversarial purification (Nie et al., 2022; Wang et al., 2022; Lee & Kim, 2023; Bai et al., 2024; Song et al., 2024; Zollicoffer et al., 2025) is a data preprocessing technique aimed at transforming adversarial images back into their original clean images during the testing phase. Compared to adversarial training (Gosch et al., 2023; Wang et al., 2024; Singh et al., 2024), it offers advantages such as decoupling training and testing, and strong generalization. The main challenge faced by adversarial purification is how to eliminate adversarial perturbations while preserving the original semantic information as much as possible. Therefore, it is essential to explore how adversarial perturbations damage the images.

Current research (Chen et al., 2022; Wang et al., 2020; Han et al., 2021; Maiya et al., 2021; Han et al., 2021; Maiya et al., 2023) studies the distribution of adversarial perturbations in the frequency domain, but they lack quantitative analysis and typically do not distinguish between amplitude spectrum and phase spectrum. Due to the different distribution characteristics of amplitude spectrum and phase spectrum, in this paper, we further decompose images into amplitude spectrum and phase spectrum. We statistically analyze the variations in amplitude spectrum and phase spectrum of original clean images and adversarial images across multiple models, attack methods, and different perturbation radii (more experimental results and details in the Appendix C). From Figure 1 we find that for both the amplitude spectrum and the phase spectrum, the degradation caused by adversarial perturbations exhibits an approximately monotonically increasing trend with frequency.

Existing methods of diffusion-based adversarial purification attempt to drown adversarial perturbations as part of isotropic noise through the forward process, and then recover clean images through the reverse process. However, we theoretically prove that this strategy will destroy all fre-

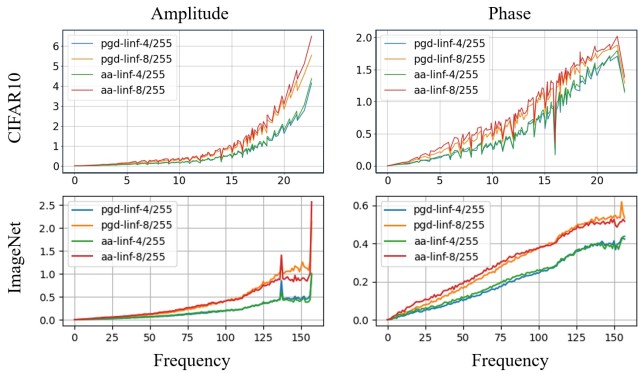

*Figure 1.* We decompose the image into the amplitude spectrum (left) and the phase spectrum (right), and calculate the differences between the adversarial images and the original images, respectively. The damage caused by adversarial perturbations tends to increase monotonically with frequency for both spectra.

quency components in both the amplitude spectrum and phase spectrum and this destruction becomes increasingly severe as the time-step $t$ increases. The empirical findings above suggest that if we want to restore adversarial images to their original clean images, we should minimize the disruption to the low-frequency amplitude spectrum and phase spectrum to ensure consistency and preserve the inherent characteristics of the original images. Therefore, we believe that current diffusion-based adversarial purification methods cause excessive damage to the semantic information of the input images.

Motivated by experimental findings and theoretical analysis, we propose a novel adversarial purification method that can remove adversarial perturbations from adversarial images while minimizing destruction to the image. Specifically, at each time-step during the reverse process, we decompose the predicted estimated clean image into its amplitude spectrum and phase spectrum. For amplitude spectrum, since low-frequency components of the adversarial image are almost unaffected, we replace the low-frequency part of the estimated image's amplitude spectrum with the low-frequency part of the adversarial image's amplitude spectrum. For phase spectrum, we project the estimated image's low-frequency phase spectrum onto a certain range of the adversarial image's low-frequency phase spectrum. This is because the low-frequency phase spectrum is less affected by adversarial perturbations, allowing us to extract coarse-grained structural information from the image while gradually aligning it with the low-frequency phase information of natural images. Our proposed method selectively retains low-frequency phase and amplitude spectrum information, which not only preserves some structural and content information of the image but also provides prior information for the restoration of high-frequency details.

Overall, the contribution of this paper is as follows:

1. We decompose the image into amplitude spectrum and phase spectrum, and explore how adversarial perturbation disturbs the original image from the perspective of the frequency domain.

2. We theoretically demonstrate that current diffusion-based purification methods excessively destroy the amplitude spectrum and phase spectrum of input images.

3. Our proposed method retains the original structural information and content while eliminating adversarial perturbations by selectively preserving the amplitude spectrum and phase spectrum at each time-step.

4. Extensive experiments show that our method outperforms other methods by a promising improvement against various adversarial attacks.

## 2. Related Work

Adversarial purification is a technique designed to eliminate adversarial perturbations from input images before classification, ensuring more robust model performance. These purification approaches can be categorized into two primary paradigms: training-based methods that necessitate dataset preparation, and diffusion-based techniques that offer a training-agnostic solution, capable of operating without direct access to original training data.

### 2.1. Training-Based Adversarial Purification

(Song et al., 2018) empirically demonstrates that adversarial examples predominantly exist in the low probability areas of the training distribution and aims to redirect them back towards this distribution. (Samangouei et al., 2018) initially models the distribution of clean images and subsequently seeks the nearest clean sample to the adversarial example during inference. (Naseer et al., 2020) generates perturbed images using a self-supervised perturbation attack that disrupts the deep perceptual features and projects back the perturbed images close to the perceptual space of clean images. (Zhou et al., 2021) introduces a method to learn generalizable invariant features across various attacks using an encoder, engaging in a zero-sum game to reconstruct the original image with a decoder. (Lin et al., 2024) combines adversarial training and purification techniques via employing random transforms to disrupt adversarial perturbations and fine-tunes a purifier model using adversarial loss. (Tang & Zhang, 2024) leverages the phenomenon of FGSM robust overfitting to enhance the robustness of deep neural networks against unknown adversarial attacks. Nonetheless, these approaches necessitate training on the training dataset, which is time-intensive and lacks generalizability.

## 2.2. Diffusion-based Adversarial Purification

(Yoon et al., 2021) shows that an energy-based model trained with denoising score-matching can quickly purify attacked images within a few steps. (Nie et al., 2022) begins by adding a small amount of noise to the images and then recovers the clean image through a reverse generative process (Wang et al., 2022) suggests using the adversarial image as a reference during the reverse process, which helps ensure that the purified image aligns closely with the original clean image. (Lee & Kim, 2023) proposed a new gradient estimation method and introduced a stepwise noise scheduling strategy to enhance the effectiveness of the current purification methods. (Song et al., 2024) proposes a method that reduces the negative impact of adversarial perturbations by mimicking the generative process with clean images as input. (Bai et al., 2024) designs the forward process with the proper amount of Gaussian noise added and the reverse process with the gradient of contrastive loss as the guidance of diffusion models for adversarial purification. However, our theoretical analysis demonstrates that these methods excessively disrupt the semantic information of the images.

## 3. Theoretical Study

We perform discrete Fourier transform (DCT) on the input image $\mathbf{x}_0$ and the noisy image $\mathbf{x}_t$ obtained using forward process (Ho et al., 2020) at time-step $t$ as follows:

$$
\begin{aligned}
\mathbf{x}_0(u,v) = DCT(\mathbf{x}_0) = |\mathbf{x}_0(u,v)|e^{i\phi_{\mathbf{x}_0}(u,v)}, \\
\mathbf{x}_t(u,v) = DCT(\mathbf{x}_t) = |\mathbf{x}_t(u,v)|e^{i\phi_{\mathbf{x}_t}(u,v)},
\end{aligned} \tag{1}
$$

where $u$ and $v$ are coordinates at frequency domain. $|\mathbf{x}_0(u,v)|$ and $|\mathbf{x}_t(u,v)|$ denote the amplitude spectrum, $\phi_{\mathbf{x}_0}(u,v)$ and $\phi_{\mathbf{x}_t}(u,v)$ denote the phase spectrum.

**Definition 3.1.** (Difference of amplitude) Given the amplitude $|\mathbf{x}_0(u,v)|$ of the input image $\mathbf{x}_0$ and the amplitude $|\mathbf{x}_t(u,v)|$ of the noisy image $\mathbf{x}_t$, respectively. We definite the difference between there two amplitude of the images at arbitrary coordinate $(u,v)$ are as follows:

$$
\Delta A_t(u,v) = |\mathbf{x}_t(u,v)| - |\mathbf{x}_0(u,v)|. \tag{2}
$$

This value represents the degree of variation in the image content.

**Theorem 3.2.** *(Proof in Appendix D.1) The variance of the difference of amplitude at time-step $t$ between clean image $\mathbf{x}_0$ and noisy image $\mathbf{x}_t$ at arbitrary coordinates $(u,v)$ at frequency domain is as follows:*

$$
Var(\Delta A_t(u,v)) \approx \frac{1-\overline{\alpha}_t}{2} - \frac{(1-\overline{\alpha}_t)^2}{16|\mathbf{x}_0(u,v)|\overline{\alpha}_t}. \tag{3}
$$

*The RHS is monotonically increasing with respect to $t$, This means that as $t$ increases, the amplitude spectrum of the original image at arbitrary coordinate $(u,v)$ is increasingly disrupted by noise.*

**Definition 3.3.** (Difference of phase) Given the phase $\phi_{\mathbf{x}_0}$ of the input image $\mathbf{x}_0$ and the phase $\phi_{\mathbf{x}_t}$ of the noisy image $\mathbf{x}_t$, respectively. We definite the difference between there two phase of the images at arbitrary coordinate $(u,v)$ are as follows:

$$
\Delta\theta_t(u,v) = \phi_{\mathbf{x}_t}(u,v) - \phi_{\mathbf{x}_0}(u,v). \tag{4}
$$

This value represents the degree of variation in the shape and structure of the image.

**Theorem 3.4.** *(Proof in Appendix D.2) The variance of the first-order approximation of the difference of phase between clean image $\mathbf{x}_0$ and noisy image $\mathbf{x}_t$ at arbitrary coordinates $(u,v)$ at frequency domain is as follows:*

$$
Var(\Delta\theta_t(u,v)) = \frac{1}{\sqrt{1 - \frac{1}{SNR_t^2(u,v)}}} - 1, \tag{5}
$$

*where signal to noise ratio (SNR) at time-step $t$ is defined as follows:*

$$
SNR_t(u,v) = \frac{\sqrt{\overline{\alpha}_t}|\mathbf{x}_0(u,v)|}{\sqrt{1-\overline{\alpha}_t}|\epsilon(u,v)|}. \tag{6}
$$

*Obviously, $SNR_t(u,v)$ decreases monotonically with $t$, so the variance of the difference of phase $Var(\Delta\theta(u,v))$ increases monotonically with $t$. This means that as $t$ increases, the phase spectrum of the original image is increasingly disrupted by noise.*

**Remark 3.5.** From Theorem 3.2 and Theorem 3.4, we can conclude that all frequency components of the image are disrupted by the forward process, and the degree of disruption increases monotonically with $t$ ($\overline{\alpha}_t$ decreases monotonically with $t$ ). From Figure 1, we can find that as $t$ increases, while the adversarial perturbations are drowned out into part of isotropic noise, the normal content and structural information contained in the amplitude and phase spectra of low frequencies will also be disrupted by the forward process, which means that the existing method may result in excessive disruption of the input images.

## 4. Methodology

To minimize the damage to semantic information while eliminating adversarial perturbations, we propose a novel adversarial purification method named FreqPure from the perspective of the frequency domain. The core of the FreqPure method lies in selectively preserving low-frequency phase and amplitude spectrum information which are less affected by adversarial perturbations. Our approach not only maintains some structural features and content of the original image during the reverse process but also provides valuable prior constraints for the reconstruction of high-frequency information in the image.

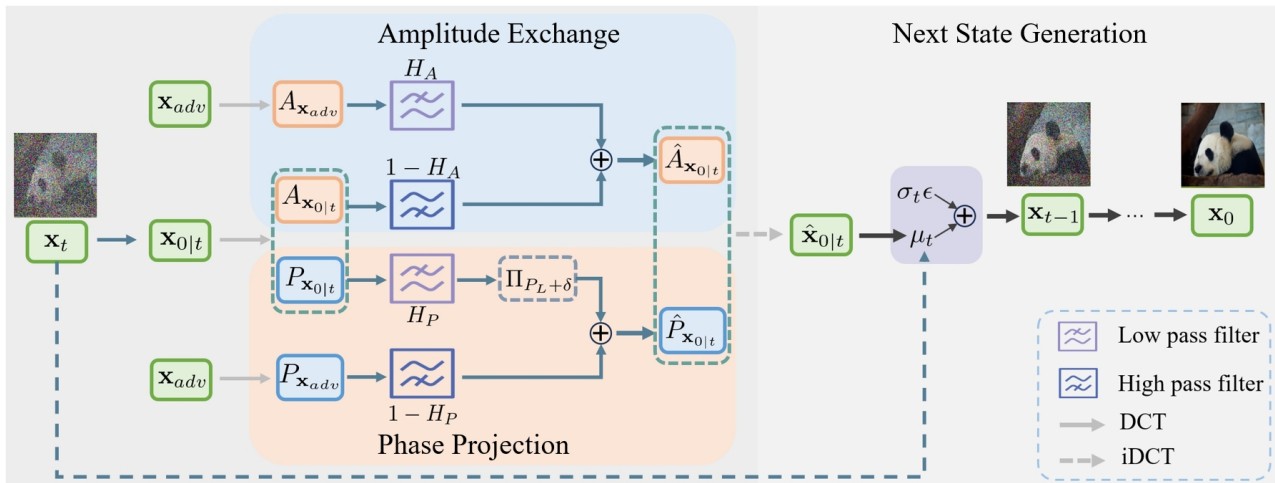

Figure 2. Pipeline of our method. The core of our method is to preserve, at each time-step of the reverse process, the amplitude and phase spectrum information of the original clean samples extracted from adversarial images as a prior. This ensures the retention of the original image's content and structural information while also providing guidance for the restoration of high-frequency details.

Given an adversarial example $\mathbf{x}_{adv}$ at time-step $t = 0$, *i.e.*, $\mathbf{x}_0 = \mathbf{x}_{adv} \in \mathbb{R}^{H \times W \times C}$. We first diffuse it according to the forward process from $t = 0$ to $t = t^*$. Then, during the reverse diffusion process at each time-step, we aim to yield clean intermediate states for refinement. Following (Wang et al., 2023), to obtain "clean" samples, we estimate $\mathbf{x}_0$ from $\mathbf{x}_t$ and the predicted noise $\mathcal{Z}_\theta(\mathbf{x}_t, t)$. We denote the estimated image $\mathbf{x}_0$ at time-step $t$ as $\mathbf{x}_{0|t}$, which can be formulated as

$$\mathbf{x}_{0|t} = \frac{1}{\sqrt{\bar{\alpha}_t}} \left( \mathbf{x}_t - \mathcal{Z}_\theta(\mathbf{x}_t, t)\sqrt{1 - \bar{\alpha}_t} \right), \qquad (7)$$

where $\mathcal{Z}_\theta$ is a neural network used by DDPM(Ho et al., 2020) to predict the noise $\epsilon$ for each time-step $t$, i.e., $\epsilon_t = \mathcal{Z}_\theta(\mathbf{x}_t, t)$. Then, we perform DCT on both the input image $\mathbf{x}_0$ and the estimated image $\mathbf{x}_{0|t}$ to decompose them into the amplitude spectrum and phase spectrum as follows:

$$(A_{\mathbf{x}_0}(u, v), P_{\mathbf{x}_0}(u, v)) = DCT(\mathbf{x}_0), \qquad (8)$$

$$(A_{\mathbf{x}_{0|t}}(u, v), P_{\mathbf{x}_{0|t}}(u, v)) = DCT(\mathbf{x}_{0|t}), \qquad (9)$$

where $A_{\mathbf{x}_0}(u, v), A_{\mathbf{x}_{0|t}}(u, v), P_{\mathbf{x}_0}(u, v)$ and $P_{\mathbf{x}_{0|t}}(u, v)$ represent the amplitude spectrum and phase spectrum of input image $\mathbf{x}_0$ and estimated image $\mathbf{x}_{0|t}$, respectively. $(u, v)$ denotes the coordinate at the frequency domain. The amplitude spectrum reflects the energy distribution of various frequency components in the image. The phase spectrum contains the structural and shape information of the image.

### 4.1. Amplitude Spectrum Exchange

Experimental reveals that low-frequency amplitude spectrum components demonstrate significant robustness to adversarial perturbations, being almost unaffected by such

disturbances. Additionally, natural signals (such as images) generally exhibit low-pass characteristics, meaning that the low-frequency power spectrum components are relatively large. Therefore, we opt to retain this portion of the amplitude spectrum and replace the low-frequency amplitude spectrum components of the estimated image $\mathbf{x}_{0|t}$ accordingly at each time-step. We first construct a filter $H_A(u, v)$ for amplitude spectrum as follows:

$$H_A(u, v) = \begin{cases} 1, & D(u, v) < D_A \\ 0, & D(u, v) > D_A \end{cases}, \qquad (10)$$

where $D_A$ is hyper-parameter, $D(u, v)$ represents the distance from point $(u, v)$ to the center of the $H \times W$ frequency rectangle in the frequency domain, which is also the magnitude of the frequency. The formula is as follows:

$$D(u, v) = [(u - H/2)^2 + (v - W/2)^2]^{\frac{1}{2}}, \qquad (11)$$

With the filter $H_A$ defined above, we can replace the low-frequency components of the estimated image's amplitude spectrum with the low-frequency components of the input sample's amplitude spectrum for each channel (Color images are typically composed of three channels: RGB) as follows:

$$\hat{A}_{\mathbf{x}_{0|t}} = A_{\mathbf{x}_{0|t}} \times (1 - H_A) + A_{\mathbf{x}_0} \times H_A, \qquad (12)$$

$\hat{A}_{\mathbf{x}_{0|t}}$ represents the updated amplitude spectrum of the estimated image $\mathbf{x}_{0|t}$.

### 4.2. Phase Spectrum Projection

Different from the amplitude spectrum, the phase spectrum is affected by adversarial perturbations at all frequency com-

ponents. Directly retaining the low-frequency phase spectrum, which is less disturbed, will preserve the adversarial perturbations while also affecting the restoration of the high-frequency phase spectrum. Therefore, we choose to project the estimated image's low-frequency phase spectrum into a certain range of the input image's low-frequency phase spectrum. First, We construct a filter $H_P$ for phase spectrum which is the same as (10) but with a different hyper-parameter $D_P$. Then, we update the phase spectrum of the sampled result as follows:

$$\hat{P}_{\mathbf{x}_{0|t}} = \Pi_{P_L+\delta}(\underbrace{P_{\mathbf{x}_0} \times H_P}_{P_L}) + P_{\mathbf{x}_{0|t}} \times (1 - H_P), \quad (13)$$

where $\Pi$ is the projection operation. $P_L$ is the low-frequency phase spectrum and $\delta$ denotes the range within which we allow variations in the low-frequency phase spectrum of the estimated image $\mathbf{x}_{0|t}$.

This strategy can benefit from two aspects: 1) We can extract coarse-grained low-frequency structural information while allowing it to vary within a certain range, enabling it to gradually align with the low-frequency structural information of natural images. 2) The coarse-grained low-frequency structural information can provide prior guidance for the recovery of high-frequency information.

Overall, we aim to extract and retain information from the original clean image by focusing on frequencies that are less affected by adversarial perturbations, thereby maximizing the preservation of the original image's structure and content, while also providing a correct prior for the restoration of high-frequency information.

### 4.3. Next State Generation

With the updated amplitude spectrum $\hat{A}_{\mathbf{x}_{0|t}}$ and phase spectrum $\hat{P}_{\mathbf{x}_{0|t}}$. We combine $\hat{A}_{\mathbf{x}_{0|t}}$ and $\hat{P}_{\mathbf{x}_{0|t}}$ obtain their representation in the time domain through the inverse discrete Fourier transformation ($iDCT$) as follows:

$$\hat{\mathbf{x}}_{0|t} = iDCT(\hat{A}_{\mathbf{x}_{0|t}}, \hat{P}_{\mathbf{x}_{0|t}}). \quad (14)$$

As suggested in (Wang et al., 2023), the next state $\mathbf{x}_{t-1}$ can be sampled from a joint distribution, which is formulated as:

$$p_\theta(\mathbf{x}_{t-1}|\mathbf{x}_t, \hat{\mathbf{x}}_{0|t}) = \mathcal{N}(\mu_t(\mathbf{x}_t, \hat{\mathbf{x}}_{0|t}); \sigma_t^2 I), \quad (15)$$

where $\mu_t(\mathbf{x}_t, \hat{\mathbf{x}}_{0|t}) = \frac{\sqrt{\alpha_{t-1}\beta_t}}{1-\alpha_t}\hat{\mathbf{x}}_{0|t} + \frac{\sqrt{\alpha_t(1-\bar{\alpha}_{t-1})}}{1-\alpha_t}\mathbf{x}_t$ and $\sigma_t^2 = \frac{1-\bar{\alpha}_{t-1}}{1-\alpha_t}\beta_t$. By using the low-frequency priors of the phase spectrum and amplitude spectrum provided by the input image to guide the sampling process in each time-step $t$, we ultimately obtain the clean image $\hat{x}_0$ with natural amplitude spectrum and phase spectrum. The complete algorithm process can refer to 4.3.

---

**Algorithm 1** Sampling Process

**Require:** Sample $\mathbf{x}_{adv}$, timestep $t^*$.
1: $\mathbf{x}_t \sim \mathcal{N}(0, I)$
2: **for** $t = t^*, \dots, 1$ **do**
3:     $\mathbf{x}_{0|t} = \frac{1}{\sqrt{\alpha_t}}\left(\mathbf{x}_t - \mathcal{Z}_\theta(\mathbf{x}_t, t)\sqrt{1-\bar{\alpha}_t}\right)$
4:     $(A_{\mathbf{x}_{0|t}}, P_{\mathbf{x}_{0|t}}) = DCT(\mathbf{x}_{0|t})$
5:     $(A_{\mathbf{x}_{adv}}, P_{\mathbf{x}_{adv}}) = DCT(\mathbf{x}_{adv})$
6:     // Amplitude Spectrum Exchange
7:     $\hat{A}_{\mathbf{x}_{0|t}} = A_{\mathbf{x}_{0|t}} \times (1 - H_A) + A_{\mathbf{x}_0} \times H_A$
8:     // Phase Spectrum Projection
9:     $\hat{P}_{\mathbf{x}_{0|t}} = \Pi_{P_L+\delta}(P_L) + P_{\mathbf{x}_{0|t}} \times (1 - H_P)$
10:     // Next State Generation
11:     $\hat{\mathbf{x}}_{0|t} = iDCT(\hat{A}_{\mathbf{x}_{0|t}}, \hat{P}_{\mathbf{x}_{0|t}})$
12:     $\mathbf{x}_{t-1} \sim p(\mathbf{x}_{t-1}|\mathbf{x}_t, \hat{\mathbf{x}}_{0|t})$
13: **end for**
14: **Output** $\mathbf{x}_0$

---

## 5. Experiment

### 5.1. Experimental Settings

**Datasets and network architectures.** Three datasets are utilized for evaluation: CIFAR-10 (Krizhevsky & Hinton, 2009), SVHN and ImageNet (Deng et al., 2009). Our results are compared against several prominent defense techniques listed in the standardized benchmark RobustBench (Croce et al., 2021) for both CIFAR-10 and ImageNet, and we also examine various adversarial purification methods. For CIFAR-10, we employ two widely used classifier architectures: WideResNet-28-10 and WideResNet-70-16 (Zagoruyko & Komodakis, 2016). In the case of SVHN, WideResNet-28-10 acts as the backbone, whereas ResNet-50 (He et al., 2016) is utilized for ImageNet.

**Adversarial attack methods.** We evaluate strong adaptive attacks (Athalye et al., 2018; Tramer et al., 2020) against our approach and other adversarial purification methods. The well-known AutoAttack (Croce & Hein, 2020) is implemented under $\ell_\infty$ and $\ell_2$ threat models. Furthermore, the projected gradient descent (PGD) attack (Madry et al., 2018) is assessed on our method, as suggested in (Lee & Kim, 2023). To account for the randomness introduced by the diffusion and denoising processes, Expectation Over Transformation (EOT) (Athalye et al., 2018) is adapted for these adaptive attacks. Additionally, we employ the BPDA+EOT (Hill et al., 2021) attack to facilitate a fair comparison with other adversarial purification methods. Lastly, following the recommendations of (Lee & Kim, 2023), a surrogate process is utilized to derive the gradient of the reverse process for white-box attacks.

**Pre-trained Models.** We utilize the unconditional CIFAR-10 checkpoint of EDM supplied by NVIDIA (Karras et al., 2022) for the CIFAR-10 dataset. For the ImageNet experiments, we adopt the 256x256 unconditional diffusion check-

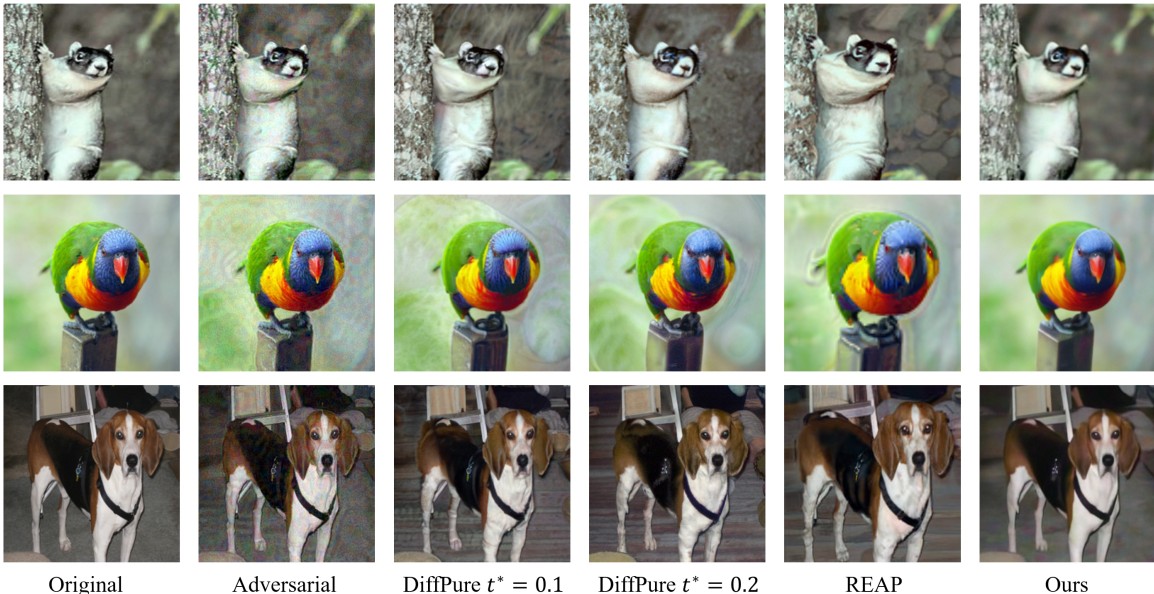

|  |  |  |  |  |  |
|---|---|---|---|---|---|
| Original | Adversarial | DiffPure $t^* = 0.1$ | DiffPure $t^* = 0.2$ | REAP | Ours |

*Figure 3.* Visualization of origianl clean images , adversarial images and purified images. The images purified by our method are most similar to the origianl clean images.

point from the guided-diffusion library. The pre-trained classifier for CIFAR-10 is obtained from RobustBench (Croce et al., 2021), whereas the classifier weights for ImageNet are sourced from the TorchVision library.

**Evaluation metrics.** To evaluate the effectiveness of defense methods, we employ two metrics: standard accuracy, which is calculated on clean images, and robust accuracy, assessed on adversarial examples. Given the significant computational expense associated with evaluating models against adaptive attacks, we randomly sample a fixed subset of 512 images from the test set for robust evaluation, consistent with (Nie et al., 2022; Lee & Kim, 2023; Song et al., 2024; Lin et al., 2024; Bai et al., 2024). In all experiments, we report the mean and standard deviation across three runs to evaluate both standard and robust accuracy.

**Implementation details.** We adhere to the configurations outlined in (Lee & Kim, 2023). Diffusion-based purification methods are evaluated using the PGD attack with 200 update iterations, while BPDA and AutoAttack are assessed with 100 update iterations, except for ImageNet, which utilizes 20 iterations. The number of EOT is set to 20, and the step size is 0.007. For randomized defenses, such as those in (Nie et al., 2022; Lee & Kim, 2023; Song et al., 2024; Bai et al., 2024), we employ the random version of AutoAttack, whereas the standard version is used for static defenses.

### 5.2. Experimental Results

**CIFAR10** We perform extensive experiments on the CIFAR-10 dataset, comparing our method with other approaches using two model architectures: WideResNet-28-

*Table 1.* Standard and robust accuracy of different Adversarial Training (AT) and Adversarial Purification (AP) methods against PGD and AutoAttack $\ell_\infty(\epsilon = 8/255)$ on CIFAR-10. $^*$ utilizes half number of iterations for the attack due to the high computational cost. $^\dagger$ indicates the requirement of extra data. The result with an underline indicates the second highest.

| Type | Method | Standard Acc. | Robust Acc. PGD | Robust Acc. AutoAttack |
|---|---|---|---|---|
| | | WideResNet-28-10 | | |
| AT | (Gowal et al., 2021) | 88.54 | 65.93 | 63.38 |
| | (Gowal et al., 2020)$^\dagger$ | 87.51 | 66.01 | 62.76 |
| | (Pang et al., 2022) | 88.62 | 64.95 | 61.04 |
| AP | (Yoon et al., 2021) | 85.66±0.51 | 33.48±0.86 | 59.53±0.87 |
| | (Nie et al., 2022) | 90.07±0.97 | 56.84±0.59 | 63.60±0.81 |
| | (Lee & Kim, 2023) | 90.16±0.64 | 55.82±0.59 | 70.47±1.53 |
| | (Bai et al., 2024) | 91.41 | 49.22$^*$ | 77.08 |
| | (Zollicoffer et al., 2025) | 84.20 | - | 59.14 |
| | (Lin et al., 2024) | 90.62 | - | 72.85 |
| | Ours | **92.19 ±0.33** | **59.39±0.79** | **77.35±2.14** |

10 and WideResNet-70-16. Robust accuracy is evaluated under three types of attacks: BPDA+EOT, PGD, and AutoAttack. For (Gowal et al., 2021; 2020; Pang et al., 2022), they typically construct adversarial examples based on other attack methods for adversarial training. Focusing on the $\ell_\infty$ attack, as illustrated in Table 1, our method demonstrates a significant advantage over other baselines when using WideResNet-28-10 as the backbone. Our method not only improves the standard accuracy metric by 0.78% but also enhances robust accuracy under PGD and AutoAttack attacks by 2.55% and 0.27%, respectively. When WideResNet-70-16 is used as the backbone, from

*Table 2.* Standard and robust accuracy of different Adversarial Training (AT) and Adversarial Purification (AP) methods against PGD and AutoAttack $\ell_\infty(\epsilon = 8/255)$ on CIFAR-10. * The number of iterations for the attack is half that of the other methods for less computational overhead. † indicates the requirement of extra data. The result with an underline indicates the second highest.

| Type | Method | Standard Acc. | Robust Acc. | |
|---|---|---|---|---|
| | | | PGD | AutoAttack |
| | WideResNet-70-16 | | | |
| AT | (Rebuffi et al., 2021)† | 92.22 | 69.97 | 66.56 |
| | (Gowal et al., 2020)† | 91.10 | 68.66 | 66.10 |
| | (Gowal et al., 2021) | 88.75 | 69.03 | 65.87 |
| AP | (Yoon et al., 2021) | 86.76±1.15 | 37.11±1.35 | 60.86±0.56 |
| | (Nie et al., 2022) | 90.43±0.60 | 51.13±0.87 | 66.06±1.17 |
| | (Lee & Kim, 2023) | 90.53±0.1 | 56.88±1.06 | 70.31±0.62 |
| | (Bai et al., 2024) | 92.97 | 48.83* | 79.10 |
| | (Zollicoffer et al., 2025) | 84.60 | - | 66.40 |
| | (Zollicoffer et al., 2025) | 86.90 | - | 59.20 |
| | (Lin et al., 2024) | 91.99 | - | 76.37 |
| | Ours | 92.52±0.53 | 62.50±2.73 | 78.13±1.95 |

*Table 3.* Standard and robust accuracy against PGD and AutoAttack $\ell_2(\epsilon = 0.5)$ on CIFAR-10. Adversarial Training (AT) and Adversarial Purification (AP) methods are evaluated. *The number of iterations for the attack is half that of the other methods for less computational overhead. † indicates the requirement of extra data. ‡ adopts WideResNet-34-10 as the backbone, with the same width but more layers than the default one. The result with an underline indicates the second highest.

| Type | Method | Standard Acc. | Robust Acc. | |
|---|---|---|---|---|
| | | | PGD | AutoAttack |
| | WideResNet-28-10 | | | |
| AT | (Rebuffi et al., 2021)† | 91.79 | 85.05 | 78.80 |
| | (Augustin et al., 2020)‡ | 93.96 | 86.14 | 78.79 |
| | (Pang et al., 2022)‡ | 90.93 | 83.75 | 77.24 |
| AP | (Yoon et al., 2021) | 85.66±0.51 | 73.32±0.76 | 79.57±0.38 |
| | (Nie et al., 2022) | 91.41±1.00 | 79.45±1.16 | 81.7±0.84 |
| | (Lee & Kim, 2023) | 90.16±0.64 | 83.59±0.88 | 86.48±0.38 |
| | (Bai et al., 2024) | 91.41 | 86.13* | 80.92 |
| | (Zollicoffer et al., 2025) | 84.40 | - | 77.90 |
| | (Zollicoffer et al., 2025) | 84.20 | - | 73.60 |
| | (Lin et al., 2024) | 90.62 | - | 80.47 |
| | Ours | 92.19 ±0.33 | 87.89±1.17 | 89.06±0.43 |

table 2 we can see that though our method decreases the standard accuracy metric by 0.45%, our method enhances robust accuracy under PGD and AutoAttack attacks by 5.62% and 1.76%, respectively. We also assess the accuracy of our method and other baselines against the $\ell_2$ attack. From Table 3, we observe that, When WideResNet-28-10 is used as the backbone, Our method improves standard accuracy by 0.78%, and robust accuracy increases by 1.86% and 2.68%, respectively. As shown in Table 4 when WideResNet-70-16 is the backbone, our method outperforms other baselines, though decreasing by 0.45% in standard accuracy and by 3.62% and 4.25% in robust accuracy under PGD and AutoAttack, respectively. Additionally, we apply the BPDA+EOT attack, which approximates differentiability. As shown in Table 5, Our method leads by 0.82% in Standard Accuracy and 0.64% in Robust Accuracy. To measure the degree of similarity between the purified samples and the original clean samples, We plot the distribution map of the purified images. As shown in Figure 4, our method yields a distribution of purified images that is the most similar to the original images when compared to other methods. In addition to qualitative analysis, we also conduct some quantitative analyses. Here, We opt to utilize measures like DINO similarity (Caron et al., 2021; Oquab et al., 2024) and CLIP similarity (Radford et al., 2021), which calculate the cosine similarity between embeddings extracted from two images. From Table 7, Our method achieves the highest similarity score, indicating that the images purified by our method are the most akin to the original clean images. This is attributed to our strategy of appropriately preserving the inherent content and structural information of the images during the purification process. Overall, our method outperforms others, demonstrating its effectiveness in preserving semantic information while eliminating adversarial perturbations.

*Table 4.* Standard and robust accuracy against PGD and AutoAttack $\ell_2(\epsilon = 0.5)$ on CIFAR-10. Adversarial Training (AT) and Adversarial Purification (AP) methods are evaluated. (*The number of iterations for the attack is half that of other methods for less computational overhead. † methods need extra data.) The result with an underline indicates the second highest.

| Type | Method | Standard Acc. | Robust Acc. | |
|---|---|---|---|---|
| | | | PGD | AutoAttack |
| | WideResNet-70-16 | | | |
| AT | (Rebuffi et al., 2021)† | 95.74 | 89.62 | 82.32 |
| | (Gowal et al., 2020)† | 94.74 | 88.18 | 80.53 |
| | (Rebuffi et al., 2021) | 92.41 | 86.24 | 80.42 |
| AP | (Yoon et al., 2021) | 86.76±1.15 | 75.666±1.29 | 80.43±0.42 |
| | (Nie et al., 2022) | 92.15±0.72 | 82.97±1.38 | 83.06±1.27 |
| | (Lee & Kim, 2023) | 90.53±0.14 | 83.75±0.99 | 85.59±0.61 |
| | (Bai et al., 2024) | 92.97 | 84.37* | 83.01 |
| | (Lin et al., 2024) | 91.99 | - | 81.35 |
| | Ours | 92.52±0.53 | 87.89±1.17 | 89.84±1.56 |

**ImageNet** We evaluate ResNet50 as the backbone of the ImageNet dataset under PGD attacks, consistent with (Nie et al., 2022; Lee & Kim, 2023; Bai et al., 2024). From Table 6, our method outperforms competitors by 0.53% in standard accuracy and 3.52% in robust accuracy, significantly exceeding the baseline. Additionally, the method from (Nie et al., 2022) achieves a standard accuracy of 73.96% and a robust accuracy of 40.63% when $t^* = 0.2$. For $t^* = 0.3$, standard accuracy decreases to 72.85%, while robust accuracy increases to 48.24%. However, at $t^* = 0.4$, both standard accuracy and robust accuracy decline to 61.71% and 41.67%, respectively. This demonstrates that traditional methods increasingly destroy semantic information as $t$ increases, which aligns with our theoretical proof. In addition, we also visualize the purified images. From Figure 3, we observe that compared to DiffPure and

*Table 5.* Standard and robust accuracy against BPDA+EOT $\ell_\infty(\epsilon = 8/255)$ on CIFAR-10. The result with an underline indicates the second highest.

| Method | Purification | Accuracy | |
|---|---|---|---|
| | | Standard | Robust |
| WideResNet-28-10 | | | |
| (Song et al., 2018) | Gibbs Update | 95.00 | 9.00 |
| (Yang et al., 2019) | Mask+Recon | 94.00 | 15.00 |
| (Hill et al., 2021) | EBM+LD | 84.12 | 54.90 |
| (Yoon et al., 2021) | DSM+LD | 85.66±0.51 | 66.91±1.75 |
| (Nie et al., 2022) | Diffusion | 90.07±0.97 | 81.45±1.51 |
| (Bai et al., 2024) | Diffusion | 91.37±1.21 | 85.51±0.81 |
| (Wang et al., 2022) | Diffusion | 89.96±0.40 | 75.59±1.26 |
| (Song et al., 2024) | Diffusion | 89.88±0.35 | 88.43±0.83 |
| (Lee & Kim, 2023) | Diffusion | 90.16±0.64 | 88.40±0.88 |
| Ours | Diffusion | **92.19±0.79** | **89.07±0.79** |

*Table 6.* Standard and robust accuracy against PGD $\ell_\infty(\epsilon = 4/255)$ on ImageNet. ResNet-50 is used as the classifier. The result with an underline indicates the second highest.

| Type | Method | Accuracy | |
|---|---|---|---|
| | | Standard | Robust |
| ResNet-50 | | | |
| AT | (Salman et al., 2020) | 63.86 | 39.11 |
| | (Engstrom et al., 2019) | 62.42 | 33.20 |
| | Wong et al. (Wong et al., 2020) | 53.83 | 28.04 |
| AP | ($t^* = 0.2$) (Nie et al., 2022) | 73.96 | 40.63 |
| | ($t^* = 0.3$) (Nie et al., 2022) | 72.85 | 48.24 |
| | ($t^* = 0.4$) (Nie et al., 2022) | 61.71 | 41.67 |
| | (Bai et al., 2024) | 70.41 | 41.70 |
| | (Lee & Kim, 2023) | 71.42 | 46.59 |
| | (Song et al., 2024) | 62.25 | 51.14 |
| | (Zollicoffer et al., 2025) | **73.98** | 56.54 |
| | Ours | 71.88 | **59.77** |

REAP, our method achieves the best restoration effect. This is attributed to the prior guidance provided by extracting the low-frequency phase spectrum and amplitude spectrum with smaller perturbations during the image restoration process, which allows for effective recovery of high-frequency details. Our method decreases standard accuracy by 2.1%, but improves robust accuracy by 3.23%, respectively. With average accuracy improves by 0.55%. In general, these experiments can demonstrate the effectiveness of our approach.

*Table 7.* To measure the similarity between the purified image and the clean image, we calculated the DINO similarity and CLIP similarity between them.

| | | DiffPure | REAP | CGAP | Ours |
|---|---|---|---|---|---|
| DINO | Standard | 0.887 | 0.860 | 0.879 | **0.917** |
| | Robust | 0.820 | 0.805 | 0.822 | **0.877** |
| CLIP | Standard | 0.961 | 0.952 | 0.962 | **0.972** |
| | Robust | 0.949 | 0.946 | 0.954 | **0.956** |

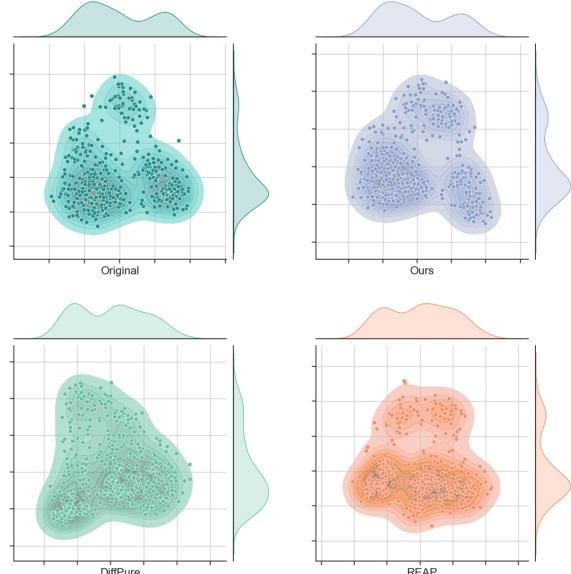

*Figure 4.* Joint distribution of the original images and purified images. The distributions of the purified images by our method and the original images are the most similar.

### 5.3. Ablation Study

To demonstrate the effectiveness of our method, we conduct comprehensive ablation experiments. We test the standard accuracy and robust accuracy. We divide our method into several parts: the first part is amplitude spectrum exchange (ASE), and the other part is phase spectrum project (PSP). The backbone we chose is WideResNet-28-10, and the attack method is AutoAttack with $\ell_\infty$ and $\epsilon = 8/255$.

After removing ASE and PSP, the method becomes the same as (Lee & Kim, 2023). However, due to the different number of time-step used in our method, there are some differences in the results compared to the original method. All ablation studies are conducted under our predefined noise time-step to verify the effectiveness of each module. From the table 8, we can see that removing either ASE or PSP will affect standard accuracy and robust accuracy, proving that our improvement is effective.

*Table 8.* Standard accuracy and robust accuracy under different combinations. WideResNet28-10 servers as the backbone.

| ASE | PSP | Standard | Robust | Average |
|---|---|---|---|---|
| ✗ | ✗ | 91.79 | 71.68 | 81.73 |
| ✓ | ✗ | 89.07 | 77.35 | 83.21 |
| ✗ | ✓ | 91.41 | 76.17 | 83.79 |
| ✓ | ✓ | **92.19** | **77.35** | **84.77** |

# 6. Conclusion

We analyze how adversarial perturbations disrupt the amplitude spectrum and phase spectrum from the perspective of the frequency domain. Additionally, we theoretically prove that current diffusion-based adversarial purification methods excessively damage images. Based on these, we propose refining the low-frequency amplitude spectrum and phase spectrum of the estimated image at each time-step during the reverse process. This approach not only preserves some structural information and image content but also guides the restoration of high-frequency components. However, both our approach and previous work still face challenges in accurately and quickly calculating gradients. This poses difficulties in evaluating the effectiveness of defense methods.

# Impact Statement

The work presented in this paper seeks to significantly advance the emerging field of machine learning security, specifically addressing vulnerabilities related to adversarial attacks. Recognizing that adversarial images pose severe threats to the integrity and reliability of machine learning models, we introduce a novel purification method leveraging diffusion models from a frequency domain perspective. Our approach uniquely exploits the frequency characteristics inherent to adversarial perturbations, enabling efficient and effective mitigation of malicious modifications to input images. This purification process enhances the robustness of model predictions, safeguarding them against a broad spectrum of adversarial strategies and ultimately improving their reliability and trustworthiness in real-world applications.

# Acknowledge

This work was supported in part by the National Science and Technology Major Project 2022ZD0119204, in part by National Natural Science Foundation of China: 62236008, 62441232, U21B2038, U23B2051, 62122075 and 62376257, in part by Youth Innovation Promotion Association CAS, in part by the Strategic Priority Research Program of the Chinese Academy of Sciences, Grant No. XDB0680201, in part by the Fundamental Research Funds for the Central Universities, in part by Xiaomi Young Talents Program.

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

## A. More Experimental Results

### A.1. Hyperparameter Sensitivity Analysis

In this section, we conduct a hyperparameter sensitivity analysis, and our method includes three hyperparameters: one for controlling the retention of the amplitude spectrum $D_A$, and the other two for controlling the retention of the phase spectrum $D_P$ and the projection range $\delta$. WideResNet28-10 serves as the classifier and we use PGD as the attack method with $\ell_\infty$ and $\epsilon = 8/255$. From Figure 5, we can see that the best performance is achieved when $D_A = 3$ and $D_P = 2, \delta = 0.2$.

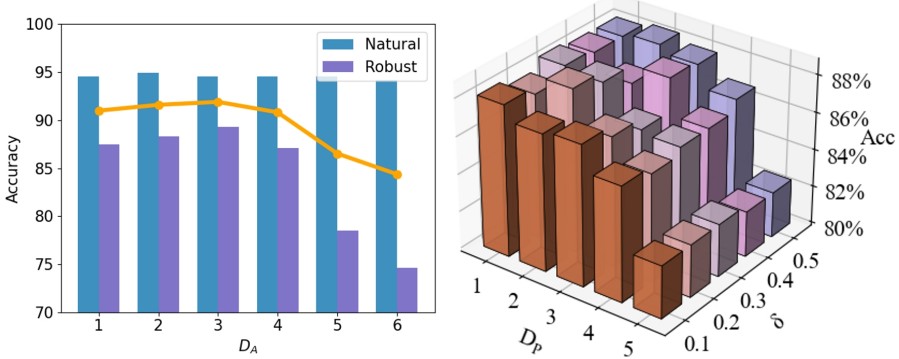

*Figure 5.* Robust and Standard Accuracy under different thresholds $D_A$ (left) and under different combinations of $D_P$ and $\delta$ (right).

### A.2. Adaptive Attack and Surrogate Process

Strong adaptive attacks (Athalye et al., 2018; Tramer et al., 2020) require computing full gradients of diffusion-based adversarial purification methods. (Nie et al., 2022) proposes to use the adjoint method to compute full gradients of the reverse generative process. The adjoint method can compute the exact gradient in theory, but in practice, the adjoint relies on the performance of the numerical solver, whose performance becomes problematic in some cases as reported by (Zhuang et al., 2020). Furthermore, the experiments conducted by (Lee & Kim, 2023) reveal that this method tends to overestimate the robustness of the defensive measures. As suggested in (Lee & Kim, 2023), they use the approximated gradient obtained from a surrogate process. The surrogate process utilizes the fact that given the total amount of noise, we can denoise the same amount of noise with different numbers of denoising steps. Therefore, instead of using the entire denoising steps, we can mimic the original denoising process with fewer function calls, whose gradients can be obtained by back-propagating the forward and denosing process directly. To investigate whether our defense method is sensitive to the varying number of denoising steps in the surrogate process, we conducted an experimental analysis.

## B. Discrete Fourier Transform

### B.1. Preliminary

Given an image $\mathbf{x} \in \mathbb{R}^{H \times W}$, we perform a two-dimensional discrete Fourier transform (DCT) on it:

$$\mathbf{x}(u,v) = DCT(\mathbf{x}(x,y)) = \sum_{x=0}^{H-1} \sum_{y=0}^{W-1} \mathbf{x}(x,y) e^{j2\pi(\frac{ux}{H} + \frac{vy}{W})}, \qquad (16)$$

where $u = 0, 1, 2, ..., H-1$ and $v = 0, 1, 2, ..., W-1$.
When the transform $\mathbf{x}(u,v)$ is known, $\mathbf{x}(x,y)$ can be obtained using the inverse discrete Fourier transform (IDCT):

$$\mathbf{x}(x,y) = IDCT(\mathbf{x}(u,v)) = \frac{1}{HW} \sum_{u=0}^{H-1} \sum_{v=0}^{W-1} \mathbf{X}(u,v) e^{j2\pi(\frac{ux}{H} + \frac{vy}{W})}, \qquad (17)$$

where $x = 0, 1, 2, ..., H-1$ and $y = 0, 1, 2, ..., W-1$.
Since the two-dimensional discrete Fourier transform is usually a complex function, it can be represented in polar coordinates:

$$\mathbf{x}(u,v) = R(u,v) + jI(u,v) = |\mathbf{x}(u,v)|e^{j\phi(u,v)}. \qquad (18)$$

The method for calculating the amplitude is as follows:

$$|\mathbf{x}(u,v)| = \sqrt{R^2(u,v) + I^2(u,v)}. \tag{19}$$

The method for calculating the phase is as follows:

$$\phi(u,v) = \arctan(\frac{I(u,v)}{R(u,v)}). \tag{20}$$

### B.2. Property

(Linearity) For any two images $\mathbf{x}$ and $\mathbf{y}$, where $a$ and $b$ are constants, then:

$$\begin{aligned} DCT(a \cdot \mathbf{x}(x,y) + b \cdot \mathbf{y}(x,y)) &= DCT(a \cdot \mathbf{x}(x,y)) + DCT(b \cdot \mathbf{y}(x,y)) \\ &= a \cdot DCT(\mathbf{x}(x,y)) + b \cdot DCT(\mathbf{y}(x,y)) \\ &= a \cdot \mathbf{x}(u,v) + b \cdot \mathbf{y}(u,v). \end{aligned} \tag{21}$$

## C. Distribution of Adversarial Perturbations in the Frequency Domain

### C.1. Experiment Settings

We decompose the images into amplitude spectrum and phase spectrum using the discrete Fourier transform, exploring how adversarial perturbations affect the amplitude spectrum and phase spectrum. Here, we randomly selected 512 images from the ImageNet dataset. We tested different attack methods, including AutoAttack (Croce & Hein, 2020), Projected Gradient Descent (PGD (Madry et al., 2018)) under $\ell_\infty$ and $\ell_2$ nrom, as well as various perturbation radii, using different models including ResNet50 (He et al., 2016), VGG19 (Simonyan & Zisserman, 2015), ViT 6 (Dosovitskiy et al., 2021), DenseNet 7 (Huang et al., 2017) and ConvNeXT 8 (Liu et al., 2022).
Given two images, one normal image $\mathbf{x}$ and one adversarial image $\mathbf{x}_{adv}$, we decompose them into amplitude spectrum and phase spectrum using the discrete Fourier transform as follows:

$$\begin{aligned} \mathbf{x}(u,v) &= DCT(\mathbf{x}) = |\mathbf{x}(u,v)|e^{i\phi_{\mathbf{x}}(u,v)}, \\ \mathbf{x}_{adv}(u,v) &= DCT(\mathbf{x}_{adv}) = |\mathbf{x}_{adv}(u,v)|e^{i\phi_{\mathbf{x}_{adv}}(u,v)}. \end{aligned} \tag{22}$$

To investigate the variation of adversarial perturbations with frequency, we calculate the differences of the amplitude spectrum and phase spectrum between the normal image and the adversarial image.
For the amplitude spectrum, the amplitude spectrum of the image exhibits low-pass characteristics with respect to frequency, specifically:

$$|\mathbf{x}(u,v)| \propto D(u,v)^{-\alpha}, \tag{23}$$

Typically, the parameter $\alpha$ is 2 or 3. Due to the power-law distribution characteristic of the amplitude spectrum, we choose to quantify the differences between the amplitude spectra of adversarial images and normal images using rhe absolute value of the percentage difference:

$$\mathbb{E}(|\frac{|\mathbf{x}_{adv}(u,v)| - |\mathbf{x}(u,v)|}{|\mathbf{x}(u,v)|}|), \tag{24}$$

The distribution of the phase spectrum is typically random and closely related to the specific content of the image. Its values range from 0 to $2\pi$. Therefore, we choose to measure the differences between the phase spectra of adversarial images and nromal images using the absolute value of the differences:

$$\mathbb{E}(|\phi_{\mathbf{x}}(u,v) - \phi_{\mathbf{x}_{adv}}(u,v)|). \tag{25}$$

### C.2. More Experiment Results

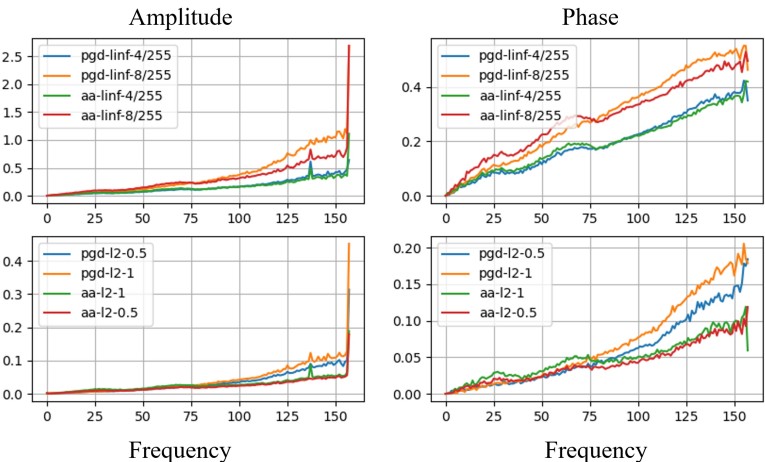

*Figure 6.* **Vision Transformer** serves as backbone. We decompose the image into the amplitude spectrum (left) and the phase spectrum (right), and calculate the differences between the adversarial images and the normal images, respectively.

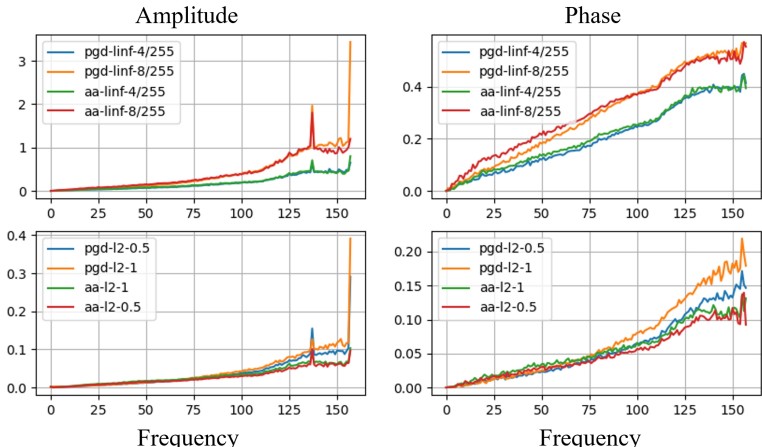

*Figure 7.* **DenseNet121** serves as backbone. We decompose the image into the amplitude spectrum (left) and the phase spectrum (right), and calculate the differences between the adversarial images and the normal images, respectively.

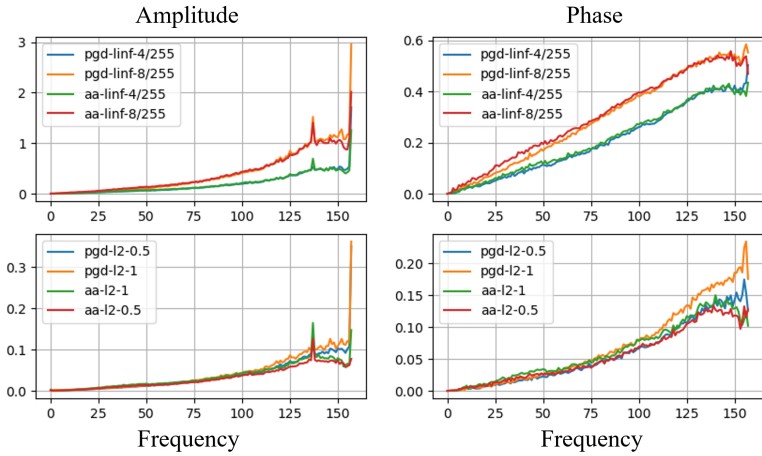

*Figure 8.* **ConvNeXT** serves as backbone. We decompose the image into the amplitude spectrum (left) and the phase spectrum (right), and calculate the differences between the adversarial images and the normal images, respectively.

# D. Proof

## D.1. Proof of Theorem 3.2.

**Theorem D.1.** (***Modified Edition***) *The variance of the difference of amplitude at time-step $t$ between clean image $\mathbf{x}_0$ and noisy image $\mathbf{x}_t$ at arbitrary coordinates $(u, v)$ at frequency domain is as follows:*

$$Var(\Delta A_t(u,v)) \approx \frac{1 - \overline{\alpha}_t}{2} - \frac{(1 - \overline{\alpha}_t)^2}{16|\mathbf{x}_0(u,v)|^2\overline{\alpha}_t}. \tag{26}$$

*The RHS is monotonically increasing with respect to $t$, This means that as $t$ increases, the amplitude spectrum of the original image at arbitrary coordinate $(u, v)$ is increasingly disrupted by noise.*

*Proof.* The forward process of DDPM (Ho et al., 2020) in the time domain is as follows:

$$\mathbf{x}_t = \sqrt{\overline{\alpha}_t}\mathbf{x}_0 + \sqrt{1 - \overline{\alpha}_t}\epsilon, \tag{27}$$

where $t$ denotes the time-step. $\mathbf{x}_0$ denotes the input clean image. $\epsilon \sim \mathcal{N}(0, \mathbf{I})$ denotes the gaussian noise. To analyze the impact of noise on different frequency components, we transform the (27) from time domain into the frequency domain via discrete Fourier transform as follows:

$$\begin{aligned} DCT(\mathbf{x}_t) &= DCT(\sqrt{\overline{\alpha}_t}\mathbf{x}_0 + \sqrt{1 - \overline{\alpha}_t}\epsilon) \\ &= \sqrt{\overline{\alpha}_t}DCT(\mathbf{x}_0) + \sqrt{1 - \overline{\alpha}_t}DCT(\epsilon), \end{aligned} \tag{28}$$

To simplify the equation. We omit the DCT and use $\mathbf{x}_0(u, v)$, $\mathbf{x}_t(u, v)$ and $\epsilon(u, v)$ at coordinate $(u, v)$ at frequency daomain:

$$\mathbf{x}_t(u, v) = \sqrt{\overline{\alpha}_t}\mathbf{x}_0(u, v) + \sqrt{1 - \overline{\alpha}_t}\epsilon(u, v), \tag{29}$$

To demonstrate the forward process will damage the amplitude spectrum, in the following we give the corresponding proof:

$$\begin{aligned} \mathbb{E}(|\mathbf{x}_t(u,v)|^2) &= \mathbb{E}(|\sqrt{\overline{\alpha}_t}\mathbf{x}_0(u,v) + \sqrt{1 - \overline{\alpha}_t}\epsilon(u,v)|^2) \\ &= \mathbb{E}(|\sqrt{\overline{\alpha}_t}\mathbf{x}_0(u,v)|^2 + |\sqrt{1 - \overline{\alpha}_t}\epsilon(u,v)|^2 + 2\Re\{\sqrt{\overline{\alpha}_t}\mathbf{x}_0(u,v) * \sqrt{1 - \overline{\alpha}_t}\epsilon(u,v)^*\}) \\ &= \mathbb{E}(\overline{\alpha}_t|\mathbf{x}_0(u,v)|^2 + (1 - \overline{\alpha}_t)|\epsilon(u,v)|^2 + 2\sqrt{\overline{\alpha}_t}\sqrt{1 - \overline{\alpha}_t}\Re\{\mathbf{x}_0(u,v) * \epsilon(u,v)^*\}), \end{aligned} \tag{30}$$

where, $\Re$ represents the real part and $\epsilon(u, v)^*$ is the complex conjugate of $\epsilon(u, v)$. The power spectral density of the noise is flat, and the noise is independent at different frequencies. Its mean is 0, and the variance is a constant (When the distribution follows a standard normal distribution, the variance is 1.):

$$\begin{aligned} \mathbb{E}(\epsilon(u, v)) &= 0 \\ \mathbb{E}(|\epsilon(u, v)|^2) &= \sigma^2 = 1 \end{aligned} \tag{31}$$

Therefore,

$$\begin{aligned} \mathbb{E}(|\mathbf{x}_t(u,v)|^2) &= \mathbb{E}(\overline{\alpha}_t|\mathbf{x}_0(u,v)|^2 + (1 - \overline{\alpha}_t)|\epsilon(u,v)|^2 + 2\sqrt{\overline{\alpha}_t}\sqrt{1 - \overline{\alpha}_t}\Re\{\mathbf{x}_0(u,v) * \epsilon(u,v)^*\}) \\ &= \overline{\alpha}_t|\mathbf{x}_0(u,v)|^2 + (1 - \overline{\alpha}_t)\mathbb{E}(|\epsilon(u,v)|^2) + 2\sqrt{\overline{\alpha}_t}\sqrt{1 - \overline{\alpha}_t}\Re\{\mathbf{x}_0(u,v) * \mathbb{E}(\epsilon(u,v)^*)\} \\ &= \overline{\alpha}_t|\mathbf{x}_0(u,v)|^2 + (1 - \overline{\alpha}_t), \end{aligned} \tag{32}$$

$$\begin{aligned} \mathbf{x}_t(u,v) = &\sqrt{\overline{\alpha}_t}\mathbf{x}_0(u,v) + \sqrt{1 - \overline{\alpha}_t}\epsilon(u,v) \\ = &\underbrace{\Re(\sqrt{\overline{\alpha}_t}\mathbf{x}_0(u,v)) + \Re(\sqrt{1 - \overline{\alpha}_t}\epsilon(u,v))}_{\sim\mathcal{N}(\Re(\sqrt{\overline{\alpha}_t}\mathbf{x}_0(u,v)), \frac{1-\overline{\alpha}_t}{2})} + i\underbrace{(\Im(\sqrt{\overline{\alpha}_t}\mathbf{x}_0(u,v)) + \Im(\sqrt{1 - \overline{\alpha}_t}\epsilon(u,v)))}_{\sim\mathcal{N}(\Im(\sqrt{\overline{\alpha}_t}\mathbf{x}_0(u,v)), \frac{1-\overline{\alpha}_t}{2})}, \end{aligned} \tag{33}$$

We can see that the means of the real part and the imaginary part are different, the variances are the same, and they are independent of each other (Richards, 2013). Therefore, the amplitude $|\mathbf{x}_t(u, v)|$ follows a Rice distribution:

$$f(|\mathbf{x}_t(u, v)|) = \frac{|\mathbf{x}_t(u, v)|}{\sigma^2} \exp\left(-\frac{|\mathbf{x}_t(u, v)|^2 + \nu^2}{2\sigma^2}\right) I_0\left(\frac{|\mathbf{x}_t(u, v)|\nu}{\sigma^2}\right), \tag{34}$$

With the assumption of $SNR_t \gg 1$, we have:

$$\mathbb{E}(|\mathbf{x}_t(u,v)|) \approx \nu \approx \nu + \frac{\sigma^2}{2\nu} = \sqrt{\overline{\alpha}_t}|\mathbf{x}_0(u,v)| + \frac{1-\overline{\alpha}_t}{4\sqrt{\overline{\alpha}_t}|\mathbf{x}_0(u,v)|}, \tag{35}$$

The variance of the difference of amplitude at time-step $t$ between clean image $\mathbf{x}_0$ and noisy image $\mathbf{x}_t$ is as follows:

$$\begin{aligned}
Var(\Delta A_t(u,v)) &= Var(|\mathbf{x}_t(u,v)| - |\mathbf{x}_0(u,v)|) \\
&= Var(|\mathbf{x}_t(u,v)|) \\
&= \mathbb{E}(|\mathbf{x}_t(u,v)|^2) - \mathbb{E}^2(|\mathbf{x}_t(u,v)|) \\
&\approx \overline{\alpha}_t|\mathbf{x}_0(u,v)|^2 + (1-\overline{\alpha}_t) - (\sqrt{\overline{\alpha}_t}|\mathbf{x}_0(u,v)| + \frac{1-\overline{\alpha}_t}{4\sqrt{\overline{\alpha}_t}|\mathbf{x}_0(u,v)|})^2 \\
&= \frac{1-\overline{\alpha}_t}{2} - \frac{(1-\overline{\alpha}_t)^2}{16|\mathbf{x}_0(u,v)|^2\overline{\alpha}_t},
\end{aligned} \tag{36}$$

When $SNR_t$ is sufficiently large, it is clearly that $Var(\Delta A_t(u,v))$ monotonically decreasing with $t$.

### D.2. Proof of Theorem 3.4.

**Theorem D.2.** *According to Theorem 3.2. in (Nie et al., 2022) that $t$ should be sufficiently small. Therefore, we assume $SNR_t > 1$. the variance of difference of phase $Var(\Delta\theta(u,v))$ between input image $\mathbf{x}_0$ and noisy image $\mathbf{x}_t$ at arbitrary coordinates $(u,v)$ at frequency domain is as follows:*

$$Var(\Delta\theta_t(u,v)) \approx \frac{1}{\sqrt{1 - \frac{1}{SNR_t^2}}} - 1, \tag{37}$$

*where signal to noise ratio (SNR) at time-step $t$ is defined as follows:*

$$SNR_t(u,v) = \frac{\sqrt{\overline{\alpha}_t}|\mathbf{x}_0(u,v)|}{\sqrt{1-\overline{\alpha}_t}|\epsilon(u,v)|}. \tag{38}$$

*Proof.* The forward process of DDPM (Ho et al., 2020) in the time domain is as follows:

$$\mathbf{x}_t = \sqrt{\overline{\alpha}_t}\mathbf{x}_0 + \sqrt{1-\overline{\alpha}_t}\epsilon. \tag{39}$$

Where $t$ denotes the time-step. $\mathbf{x}_0$ denotes the input clean image. $\epsilon \sim \mathcal{N}(0, \mathbf{I})$ denotes the gaussian noise. To analyze the impact of noise on different frequency components, we transform the (27) from time domain into the frequency domain via discrete Fourier transform as follows:

$$\begin{aligned}
DCT(\mathbf{x}_t) &= DCT(\sqrt{\overline{\alpha}_t}\mathbf{x}_0 + \sqrt{1-\overline{\alpha}_t}\epsilon) \\
&= \sqrt{\overline{\alpha}_t}DCT(\mathbf{x}_0) + \sqrt{1-\overline{\alpha}_t}DCT(\epsilon),
\end{aligned} \tag{40}$$

To simplify the equation. We omit the DCT and use $\mathbf{x}_0(u,v)$, $\mathbf{x}_t(u,v)$ and $\epsilon(u,v)$ at coordinate $(u,v)$ at frequency daomain:

$$\mathbf{x}_t(u,v) = \sqrt{\overline{\alpha}_t}\mathbf{x}_0(u,v) + \sqrt{1-\overline{\alpha}_t}\epsilon(u,v), \tag{41}$$

To demonstrate the forward process will damage the phase spectrum, in the following we give the corresponding proof:

$$\begin{aligned}
\mathbf{x}_t(u,v) &= \sqrt{\overline{\alpha}_t}\mathbf{x}_0(u,v) + \sqrt{1-\overline{\alpha}_t}\epsilon(u,v) \\
&= \sqrt{\overline{\alpha}_t}|\mathbf{x}_0(u,v)|e^{i\phi_{\mathbf{x}_0}(u,v)} + \sqrt{1-\overline{\alpha}_t}|\epsilon(u,v)|e^{i\phi_\epsilon(u,v)} \\
&= S_t e^{i\phi_{\mathbf{x}_0}(u,v)} + N_t e^{i\phi_\epsilon(u,v)} \\
&= S_t e^{i\phi_{\mathbf{x}_0}(u,v)}(1 + \frac{N_t}{S_t}e^{i(\phi_\epsilon(u,v)-\phi_{\mathbf{x}_0}(u,v))}) \\
&= S_t e^{i\phi_{\mathbf{x}_0}(u,v)}(1 + K_t e^{i\phi}).
\end{aligned} \tag{42}$$

Here we can get the difference between the phase of the original image $\mathbf{x}_0$ and the noisy image $\mathbf{x}_t$:

$$
\begin{aligned}
\Delta\theta &= \phi_{\mathbf{x}_t}(u,v) - \phi_{\mathbf{x}_0}(u,v) \\
&= \arg(1 + K_t e^{i\phi}) \\
&= \arg(1 + K_t(cos(\phi) + isin(\phi))) \\
&= \arg(\underbrace{1 + K_t\cos(\phi)}_{Real} + \underbrace{iK_t sin(\phi)}_{Imaginary}) \\
&= \arctan(\frac{K_t sin(\phi)}{1 + K_t\cos(\phi)}) \\
&\approx \frac{K_t sin(\phi)}{1 + K_t\cos(\phi)}.
\end{aligned}
\tag{43}
$$

The last line of the formula is obtained through a first-order Taylor expansion
The phase spectrum of Gaussian noise $\epsilon$ is uniformly distributed in the range $[0,2\pi]$:

$$
p(\phi_\epsilon) = \begin{cases} \frac{1}{2\pi} & \text{if } 0 \le \phi_\epsilon < 2\pi \\ 0 & \text{otherwise} \end{cases}.
\tag{44}
$$

Due to the periodicity of phase, the range of $\phi_\epsilon(u,v) - \phi_{\mathbf{x}_0}(u,v)$ is also uniformly distributed in the range $[0,2\pi]$. The expectation of the phase difference is as follows:

$$
\begin{aligned}
E(\Delta\theta) &= E(\frac{K_t sin(\phi)}{1 + K_t\cos(\phi)}) \\
&= \frac{1}{2\pi}\int_0^{2\pi} \frac{K_t sin(\phi)}{1 + K_t\cos(\phi)}d\phi \\
&= 0.
\end{aligned}
\tag{45}
$$

The Variance of the phase difference is as follows:

$$
\begin{aligned}
Var(\Delta\theta) &= E((\Delta\theta)^2) - (E(\Delta\theta))^2 = E((\Delta\theta)^2) \\
&= \frac{1}{2\pi}\int_0^{2\pi} \frac{K_t^2\sin^2(\phi)}{(1 + K_t\cos(\phi))^2}d\phi \\
&= \frac{1}{2\pi}\int_0^{2\pi} K_t\sin(\phi)d(\frac{1}{1 + K_t cos(\phi)}) \\
&= \frac{1}{2\pi}(\underbrace{\frac{K_t\sin(\phi)}{1 + K_t\cos(\phi)}\Big|_0^{2\pi}}_{0} - \int_0^{2\pi}\frac{K_t cos(\phi)}{1 + K_t\cos(\phi)}d\phi) \\
&= -\frac{1}{2\pi}\int_0^{2\pi} 1 - \frac{1}{1 + K_t\cos(\phi)}d\phi \\
&= \frac{1}{2\pi}(\int_0^{2\pi}\frac{1}{1 + K_t\cos(\phi)}d\phi - 2\pi) \\
&= \frac{1}{2\pi}(\int_{-\infty}^{+\infty}\frac{1}{1 + K_t\frac{1-t^2}{1+t^2}}\frac{2dt}{1 + t^2} - 2\pi) \\
&= \frac{1}{2\pi}(\int_{-\infty}^{+\infty}\frac{2dt}{1 + K_t + (1 - K_t)t^2} - 2\pi) \\
&= \frac{1}{2\pi}(\frac{1}{\sqrt{1 - K_t^2}}\arctan t\Big|_{-\infty}^{+\infty} - 2\pi) \\
&= \frac{1}{\sqrt{1 - K_t^2}} - 1.
\end{aligned}
\tag{46}
$$

$\square$

☐

# E. Visualization

## E.1. ImageNet

We randomly select some images for visualization and chose DiffPure (Nie et al., 2022) and REAP (Lee & Kim, 2023) as the baselines. The attack method we use here is PGD. And to make the visualization effect more pronounced, with the perturbation radius set to $\ell_\infty = 12/255$. We plot the original images, the images after adversarial attacks, and the images purified using our method and other methods.

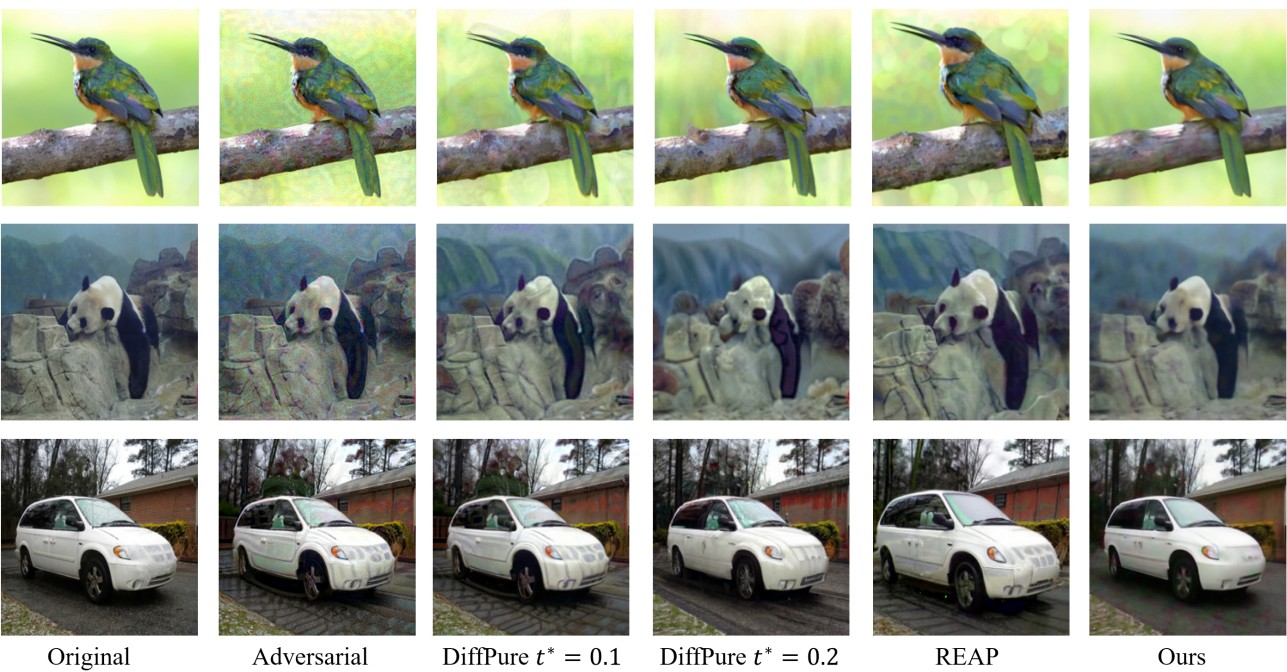

| Original | Adversarial | DiffPure $t^* = 0.1$ | DiffPure $t^* = 0.2$ | REAP | Ours |

*Figure 9.* Visualization of some randomly selected images from ImageNet dataset.

## E.2. CIFAR10

We randomly select 64 images for visualization, choosing PGD as the attack method and setting the attack radius to $\ell_\infty = 8/255$.

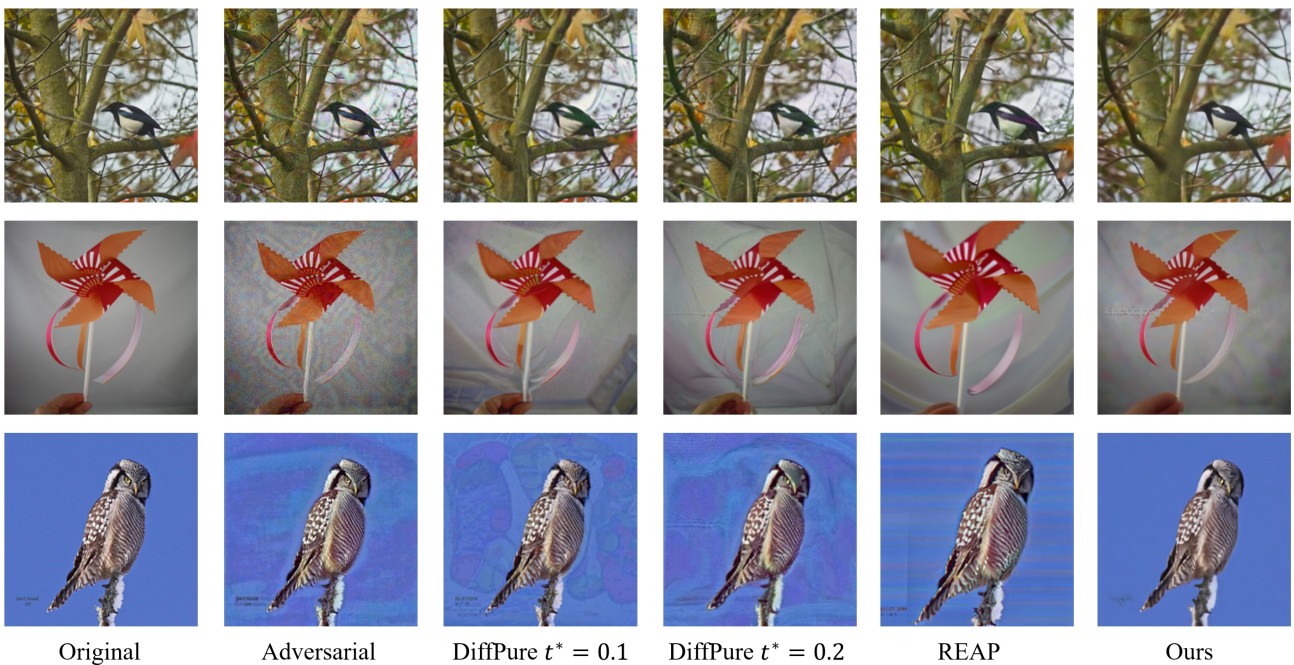

| Original | Adversarial | DiffPure $t^* = 0.1$ | DiffPure $t^* = 0.2$ | REAP | Ours |

*Figure 10.* Visualization of some randomly selected images from ImageNet dataset.

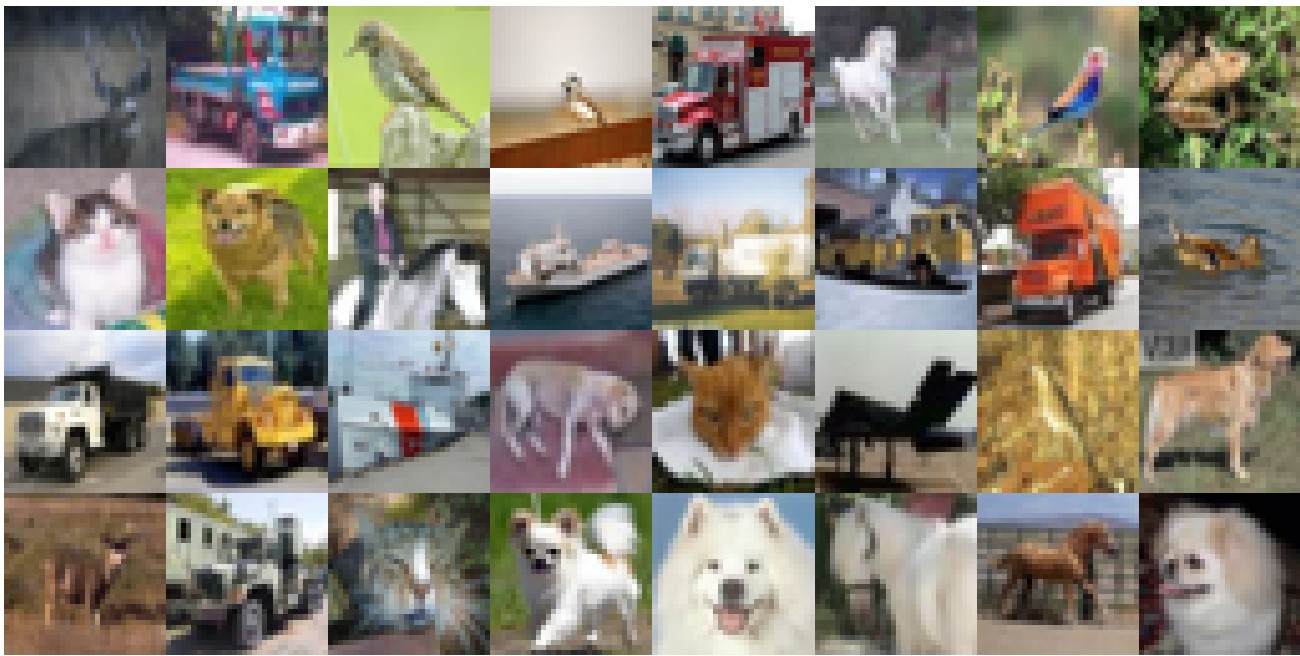

*Figure 11.* Visualization of original images randomly selected from CIFAR10 dataset.

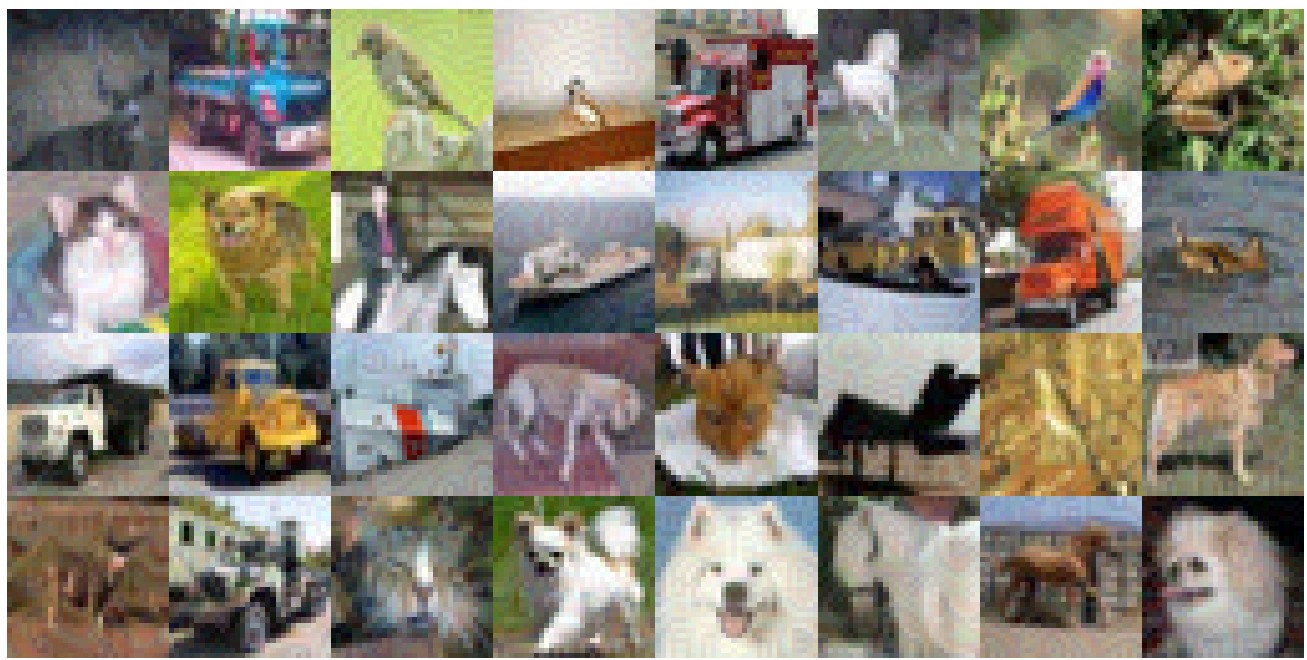

*Figure 12.* Visualization of adversarial images randomly selected from CIFAR10 dataset.

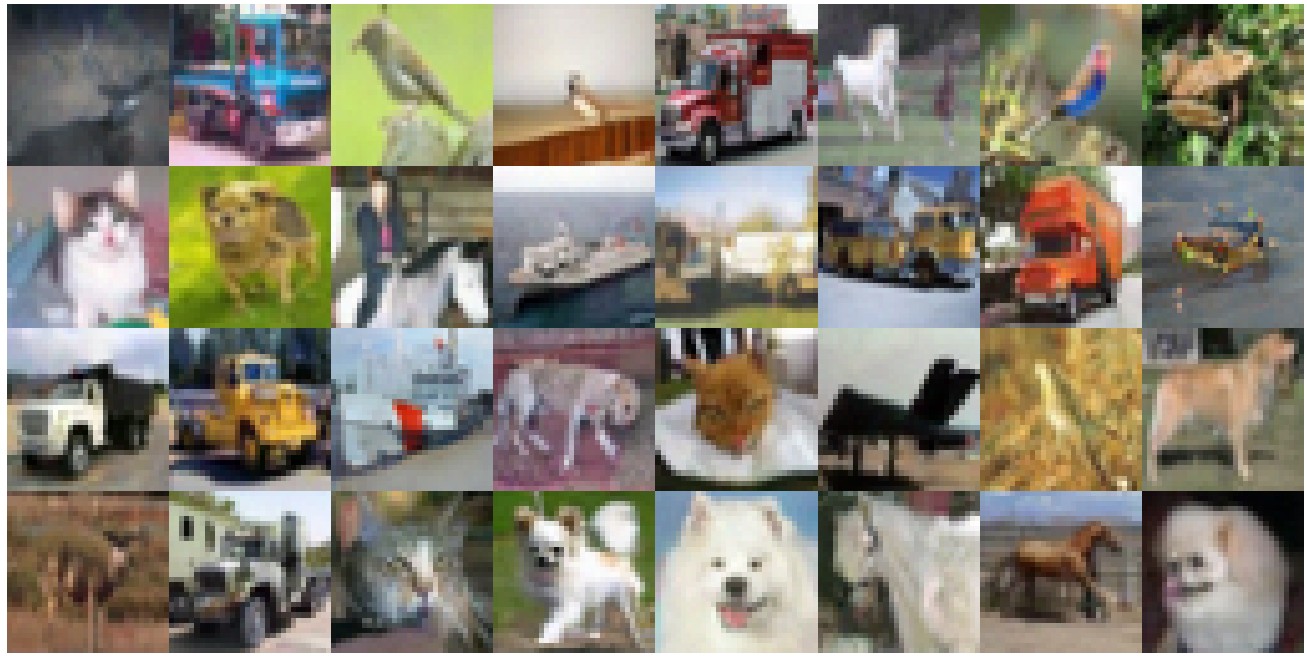

*Figure 13.* Visualization of purified images randomly selected from CIFAR10 dataset.

