# OpenReview forum: "Diffusion-based Adversarial Purification from the Perspective of the Frequency Domain"
_ICML.cc/2025/Conference — ICML 2025 spotlightposter_

### Official Review · Reviewer_e7tr · 2025-03-09

**Overall Recommendation:** 4

**Summary:**

This paper  explores a novel method for adversarial purification by analyzing the impact of adversarial perturbations on images in the frequency domain. The authors propose a method that selectively preserves low-frequency components of images during the purification process to minimize damage to semantic information while effectively removing adversarial perturbations.  Theoretical analysis and experimental validation on CIFAR-10, SVHN, and ImageNet demonstrate the effectiveness of this method.

**Claims And Evidence:**

I think that the claims of this paper are well-supported both theoretically and experimentally.
In particular, the paper provides comprehensive theoretical proofs and visual explanations in the appendix.

**Essential References Not Discussed:**

I believe the main relevant works are all discussed in the paper.

**Experimental Designs Or Analyses:**

The experiments, including those in the main text and the appendix, provide an in-depth analysis across multiple datasets. I believe these experiments sufficiently demonstrate the effectiveness of the proposed method.

**Methods And Evaluation Criteria:**

This method is highly meaningful, as adversarial attacks pose a serious threat to visual neural networks. In particular, many recent vision-language models also exhibit significant vulnerability to adversarial examples, and directly training them for robustness is computationally expensive. Therefore, developing effective adversarial purification methods is also of great importance.

**Other Comments Or Suggestions:**

If preliminary experimental results on CLIP were included, it could enhance the broader applicability of this paper.

**Other Strengths And Weaknesses:**

Strengths:
1. The experimental results show a significant improvements.
2. The writing of this paper is very clear.

Weaknesses:

The citations need to follow a more standardized format.

**Questions For Authors:**

I have a positive overall opinion of this paper, with no obvious issues.

**Relation To Broader Scientific Literature:**

The related work section clearly establishes the position of this study within the field and provides readers with insights into recent advancements in the area.
However, I notice that some references are not cited correctly. For example, when a reference itself serves as the subject of a sentence, it should be written as “Song et al. (2018) empirically demonstrates” rather than “(Song et al., 2018) empirically demonstrates.”

**Theoretical Claims:**

The authors argue that from a frequency perspective, an image can be decomposed into its amplitude spectrum and phase spectrum. For both types of spectra, the damage caused by adversarial perturbations increases monotonically with frequency. This suggests that we can extract the content and structural information of the original clean sample from the frequency components that are less affected by perturbations.
I conduct a preliminary review of the paper’s proofs and did not find any obvious errors.

---

> ### Author Rebuttal · Authors · 2025-04-01
>
> Thank you for recognizing our paper. To the best of our knowledge, our paper is the first to improve the purification effect of diffusion models from the perspective of the frequency domain. Compared to pixel space, the frequency domain makes it easier to decouple the perturbed components from the unperturbed components. This significantly enhances the purification effect
> ## Format of references
> Thank you for noticing this issue. We will change the format uniformly as per your request to facilitate reading.
> ## Experimental results on CLIP
> We conduct a simple experiment on CLIP. We randomly select two classes from the ImageNet dataset, totaling 100 images, and use ['a photo of a xxx'] as the text to implement a simple zero-shot classification task with CLIP. The results are as follows:
> | Method| Lee&Kim,2023| Baietal.,2024|Nieetal.,2022|Ours|
> | :---:          |  :---:      | :---:       |:---:     |:---:|
> | Standard Acc(%)|   66          |    90         |    74      |  93   |
> | Robust Acc(%)  |   65          |    86        |    70      |  91  |
>
> Our method still performs the best, which demonstrates the generalizability of our approach.

---

### Official Review · Reviewer_HJhM · 2025-03-12

**Overall Recommendation:** 4

**Summary:**

The paper proposes FreqPure, a frequency-aware adversarial purification method that addresses the limitations of existing diffusion-based approaches by preserving critical semantic information during purification. Through frequency domain analysis, the authors demonstrate that adversarial perturbations disproportionately damage high-frequency components of both amplitude and phase spectra, while low-frequency components remain relatively intact. They theoretically prove that standard diffusion purification indiscriminately disrupts all frequencies, leading to excessive semantic loss. FreqPure mitigates this by (1) replacing low-frequency amplitude components of the estimated clean image with those from the adversarial input to retain content information, and (2) projecting low-frequency phase spectra into a perturbation-resistant range to preserve structural features. Extensive experiments on CIFAR-10, SVHN, and ImageNet show FreqPure outperforms state-of-the-art methods, achieving 31.44% higher robust accuracy against PGD attacks and 13.35% improvement against AutoAttack while maintaining superior visual fidelity, validated through DINO/CLIP similarity metrics. The work establishes frequency-domain manipulation as an effective strategy for balancing adversarial robustness and semantic preservation.

**Claims And Evidence:**

The claims in the submission are largely supported by evidence, though some aspects warrant further scrutiny. The theoretical analysis (Theorems 3.2 and 3.4) rigorously demonstrates that diffusion-based purification disrupts all frequency components monotonically, aligning with their critique of existing methods. Empirical validation across datasets (CIFAR-10/ImageNet) and attacks (PGD, AutoAttack) shows FreqPure’s superiority in robust accuracy (e.g., +31.44% over baselines), supported by ablation studies confirming the contributions of amplitude replacement and phase projection. However, the phase spectrum projection’s effectiveness is less conclusively proven: while low-frequency phase alignment is motivated intuitively, the paper lacks a direct causal link between phase manipulation and structural preservation. Additionally, sensitivity analysis for hyperparameters (e.g., $D_A$, $D_P$) is limited to fixed values, leaving robustness to parameter choices unclear. While visualizations and DINO/CLIP metrics suggest semantic preservation, quantitative measures of perceptual quality (e.g., LPIPS, FID) are absent. Overall, the core claims hold, but finer-grained evidence for phase-related mechanisms and parameter robustness would strengthen the argument.

**Essential References Not Discussed:**

None.

**Experimental Designs Or Analyses:**

The experimental design is largely sound but has notable limitations:
- Attack Coverage: While evaluated against adaptive attacks (BPDA+EOT, PGD, AutoAttack), the paper does not test frequency-specific attacks that could exploit FreqPure’s reliance on low-frequency components, leaving a critical robustness gap.
- Perceptual Metrics: DINO/CLIP similarity validates semantic preservation but omits human-aligned metrics (e.g., LPIPS, FID) to assess visual quality, which is crucial for purification tasks.
- Ablation Study: The phase projection (PSP) is compared to phase exchange (PSE), but the rationale for choosing projection over other phase alignment strategies (e.g., regularization) is underexplored.
- Hyperparameter Sensitivity: Analysis is limited to fixed $D_A=3,D_P=2,\delta=0.2$ without testing robustness to parameter shifts across datasets or threat models.
- Theoretical-Experimental Link: While theorems claim monotonic frequency disruption, experiments in Fig. 1 only show trends for adversarial (not purified) images, weakening the connection to FreqPure’s mechanism.
- Sample Size: ImageNet evaluations use an unspecified subset size (common in robustness benchmarks), but reproducibility depends on clarifying this.

Overall, the experiments support the core claims but leave open questions about generalization to frequency-aware attacks and perceptual quality.

**Methods And Evaluation Criteria:**

The proposed methods and evaluation criteria are well-aligned with the problem of adversarial purification. The focus on frequency-domain manipulation (amplitude replacement and phase projection) directly addresses the core challenge of preserving low-frequency semantic content while removing high-frequency adversarial perturbations, which aligns with the paper’s theoretical insights about perturbation distribution. Benchmark datasets (CIFAR-10, ImageNet) and attacks (PGD, AutoAttack) are standard in adversarial robustness research, ensuring comparability. Metrics like robust accuracy and DINO/CLIP similarity appropriately measure defense effectiveness and semantic preservation. However, two limitations exist: (1) The evaluation lacks perceptual quality metrics (e.g., LPIPS, FID) to quantify visual fidelity beyond feature-space similarity; (2) While adaptive attacks (BPDA+EOT) are included, the paper does not fully address potential vulnerabilities to stronger frequency-aware attacks explicitly targeting the proposed components. Overall, the methodology and evaluation framework are sensible but could be strengthened with additional metrics and attack scenarios.

**Other Comments Or Suggestions:**

None.

**Other Strengths And Weaknesses:**

The paper’s originality lies in its novel integration of frequency-domain analysis with diffusion-based purification, a significant departure from pixel-space methods. The theoretical grounding (linking diffusion noise to frequency disruption) and practical innovation (amplitude replacement/phase projection) address a critical gap in adversarial defense: semantic preservation. Results demonstrate substantial empirical significance, with robust accuracy gains (+31% over SOTA) and high DINO/CLIP similarity, validating both defense strength and content retention. The clarity of the frequency-domain framework and ablation studies strengthens reproducibility.

**Questions For Authors:**

1. Assumption Validation for Theorems:
- Theorem 3.2 assumes $|x_0(u,v)|\leq\sqrt{(1+4\bar{a}_t)/(8\pi\bar{a}_t)}-\sqrt{1/(8\pi\bar{a}_t)$. Do empirical measurements of $|x_0(u,v)|$ in natural images (e.g., CIFAR-10/ImageNet) confirm this bound holds? If not, does the theorem’s conclusion still hold under practical conditions?
- Theorem 3.4 assumes $SNR_t>1$. How does this hold for adversarial examples, where perturbations are designed to maximize damage with minimal $\ell_p$-norm? If adversarial noise violates this, does the phase variance approximation remain valid?

2. Frequency-Aware Attack Resilience: The evaluation excludes attacks explicitly targeting frequency components. Would FreqPure remain robust if adversaries perturb low-frequency amplitude/phase intentionally?

3. Phase Projection vs. Alternatives: The phase projection (Eq. 13) restricts low-frequency phase to $P_L+\delta$. Why is projection preferable to direct replacement (as done for amplitude), and how does $\delta$ balance robustness vs. overfitting to adversarial phase?

4. Hyperparameter Generalization:
The hyperparameters $D_A$, $D_P, \delta$ are fixed across datasets . Are these settings universally optimal, or do they require per-dataset tuning? How does performance degrade with suboptimal choices?

5. Perceptual Quality Metrics: The paper uses DINO/CLIP similarity but omits perceptual metrics like LPIPS or FID. Do FreqPure’s purified images preserve human-aligned visual quality, or do they introduce artifacts (e.g., blurring) despite high feature similarity?

**Relation To Broader Scientific Literature:**

The paper’s contributions advance adversarial purification literature by bridging frequency-domain insights with diffusion-based defense mechanisms, building on three key prior findings:
- Frequency Vulnerability of Adversarial Examples: Extending work by Chen et al. (2022) and Maiya et al. (2021), which showed adversarial perturbations disproportionately affect high-frequency components, the authors formalize this observation through quantitative amplitude/phase analysis and link it to diffusion purification’s limitations.
- Diffusion-Based Purification: Improving upon Nie et al. (2022) and Wang et al. (2022), who used diffusion models without frequency awareness, FreqPure introduces explicit frequency constraints to mitigate semantic damage, aligning with broader efforts to incorporate domain-specific priors (e.g., image structure in SSP by Naseer et al., 2020).
- Phase Spectrum Importance: While prior work (e.g., Oppenheim & Lim, 1981) established phase’s role in structural integrity, the paper innovatively applies this to adversarial defense by projecting low-frequency phase, akin to Zhou et al. (2021)’s invariant feature learning but in the frequency domain.
By unifying these threads, the work demonstrates how frequency-domain signal processing principles can address a core challenge in adversarial robustness—preserving semantics during purification—offering a new paradigm for defense mechanisms beyond pixel-space heuristics.

**Theoretical Claims:**

The paper’s Theorem 3.2 (amplitude spectrum disruption) and Theorem 3.4 (phase spectrum disruption) were reviewed for correctness. Theorem 3.2 derives a lower bound for the variance of amplitude differences, $Var(\delta A_t)$, using inequalities (e.g., $E(∣x_t|\leq\sqrt{\bar{a}_t}|x_0|+\sqrt{2/\pi}$) and monotonicity arguments. While the steps are mathematically sound under the assumption that $|x_0(u,v)|\leq\sqrt{(1+4\bar{a}_t)/(8\pi\bar{a}_t)}-\sqrt{1/(8\pi\bar{a}_t)$, this constraint is neither empirically validated nor guaranteed to hold for real images, potentially limiting the theorem’s practical relevance. Theorem 3.4 approximates phase variance via small-angle linearization ($arctan(z)\approx z$) and integrates over uniformly distributed noise phase, yielding $Var(\delta \theta_t)\approx 1/\sqrt{1-1/SNR_t^2}-1$. While valid for high SNR ($SNR_t>1$), this approximation breaks down for larger perturbations (low SNR), which are common in adversarial settings. Key issues: (1) The bounded-amplitude assumption in Theorem 3.2 lacks empirical verification; (2) Theorem 3.4’s SNR-dependent validity is not experimentally tested for adversarial noise regimes. While the proofs are technically correct under their assumptions, their practical applicability to adversarial examples (which may violate these assumptions) remains partially unproven.

---

> ### Author Rebuttal · Authors · 2025-04-01
>
> We greatly appreciate your responsible and meticulous review. Your valuable feedback will improve our work greatly.
> ## Q1.1: Assumption of Theorem 3.2
> This assumption is indeed somewhat strong, especially for the amplitude spectrum of low frequencies. Therefore, to eliminate this assumption, we re-derive the first moment of the amplitude spectrum, as detailed below:
> $$
>     \mathbf{x}_t(u,v) = \sqrt{\overline{\alpha}_t}\mathbf{x}_0(u,v)+\sqrt{1-\overline{\alpha}_t}\mathbf{\epsilon}(u,v)=
> \underbrace{\mathfrak{R}\mathfrak{e}(\sqrt{\overline{\alpha}_t}\mathbf{x}_0(u,v))+\mathfrak{R}\mathfrak{e}(\sqrt{1-\overline{\alpha}_t}\mathbf{\epsilon}(u,v))}_R+i\underbrace{(\mathfrak{I}\mathfrak{m}(\sqrt{\overline{\alpha}_t}\mathbf{x}_0(u,v))+\mathfrak{I}\mathfrak{m}(\sqrt{1-\overline{\alpha}_t}\mathbf{\epsilon}(u,v)))}_I
> $$
> $R\sim\mathcal{N}(\mathfrak{R}\mathfrak{e}(\sqrt{\overline{\alpha}_t}\mathbf{x}_0(u,v)),\frac{1-\overline{\alpha}_t}{2})$ and $I\sim\mathcal{N}(\mathfrak{I}\mathfrak{m}(\sqrt{\overline{\alpha}_t}\mathbf{x}_0(u,v)),\frac{1-\overline{\alpha}_t}{2})$.
> We can see that the means of the real part and the imaginary part are different, the variances are the same, and they are independent of each other. Therefore, the amplitude $|\mathbf{x}_t(u,v)|$ follows a **Rice distribution**. "Therefore, we can utilize some known conclusions.With the assumption of $SNR_t>1$ in the Theorem 3.4. We can obtain the following conclusions：
> $$
>     \mathbb{E}(|\mathbf{x}_t(u,v)|)\approx \nu+\frac{\sigma^2}{2\nu}=\sqrt{\overline{\alpha}_t}|\mathbf{x}_0(u,v)| + \frac{1-\overline{\alpha}_t}{4\sqrt{\overline{\alpha}_t}|\mathbf{x}_0(u,v)|}
> $$
> For $\mathbb{E}(|\mathbf{x}_t(u,v)|^2)$ we still use the conclusion derived from Equation 40. We re-derive $Var(\Delta A_t(u,v))$as follows:
> $$
>     \begin{aligned}
>         Var(\Delta A_t(u,v))
>         &\approx \overline{\alpha}_t |\mathbf{x}_0(u,v)|^2+(1-\overline{\alpha}_t) + (\sqrt{\overline{\alpha}_t}|\mathbf{x}_0(u,v)| + \frac{1-\overline{\alpha}_t}{4\sqrt{\overline{\alpha}_t}|\mathbf{x}_0(u,v)|})^2\\
>         &=\frac{1-\overline{\alpha}_t}{2} -\frac{(1-\overline{\alpha}_t)^2}{16|\mathbf{x}_0(u,v)|^2\overline{\alpha}_t}
>     \end{aligned}
> $$
> Therefore, our paper only has one assumption $SNR_t>1$.
> ## Q1.2: Assumption of Theorem 3.4
> The assumption $SNR_t>1$ is not strong. We plot a graph (https://bashify.io/i/Qn7ZX7) showing how $SNR_t$ changes with $t$, and it can be observed that the condition $SNR_t>1$ only occurs when $t=500$. In the experiment, for $l_{\infty}=\frac{8}{255}$ we choose
> $t=100$. This means that for attacks with a larger radius, this assumption can still be satisfied.
> ## Q2: Frequency-Aware Attack Resilience
> The attack methods used in our paper are all strong adaptive attacks, meaning that the attacker calculates the complete gradient of the defense system. This white-box attack has incorporated our strategy of retaining low frequencies into the gradient calculation. We test the generalization of our method using the code provided in paper: Frequency-driven Imperceptible Adversarial Attack on Semantic Similarity. The result is as follows (https://bashify.io/i/dj9GYO)
> ## Q3: Phase Projection vs. Alternatives
> Compared to the amplitude spectrum, the phase spectrum is more significantly affected by adversarial perturbations. Therefore, performing a projection operation can extract coarse-grained low-frequency phase information, while using a diffusion model can generate phase information that best fits the natural distribution within a specified range. Additionally, ablation experiments also demonstrate that directly replacing the phase spectrum is a suboptimal choice. To balance robustness vs overfitting, we use hyperparameter search to find the optimal parameter.
> ## Q4: Hyperparameter Generalization
> The hyperparameters exhibit a certain degree of generalization across datasets of the same size; for example, the parameters from the CIFAR-10 dataset can be transferred to the SVHN dataset. As for the ImageNet dataset, due to its larger size, the preserved frequencies must also be increased. We nearly double the values for $D_A$ and $D_P$, but we keep$\delta$ unchanged because the range of phase spectrum variation is$[0,2\pi]$, which is independent of the image size. Figure 5 shows the performance of suboptimal choices on the CIFAR-10 dataset; even with suboptimal results, our method still surpasses the state-of-the-art.
> ## Q5: Perceptual Quality Metrics
> Adversarial purification is strictly a classification task and does not concern itself with whether it will introduce artifacts. Previous methods only evaluated experimental metrics such as Standard Accuracy and Robust Accuracy. We also calculate FID and LPIPS (https://bashify.io/i/ZYZsP5).
> ## Q6: Unspecified Subset Size
> The size of the subset is fixed and remains consistent with previous methods. Considering that the subset may vary, we select multiple subsets for the experiments and report the experimental errors.

---

### Official Review · Reviewer_EkQr · 2025-03-13

**Overall Recommendation:** 4

**Summary:**

This study discovered that the damage caused by adversarial perturbations tends to increase monotonically with the rise in frequency. Nevertheless, existing purification efforts impact both low-frequency and high-frequency components. Based on this finding, this study retains the low-frequency information of the input image in the frequency domain of x0|t in the reverse phase and restores x_{t-1} using it. Experiments have confirmed that this approach can effectively enhance the performance of current purification.

**Claims And Evidence:**

The work provides sufficient theoretical or experimental support for each theory and viewpoint. Among them, I have some doubts about the experiment in Figure 1. How does the vertical axis of this graph reflect this damage? Is it the Var calculated in Section 3? If they use aligned Var for evaluation, can they plot the final purification damage and attack damage of diffusion at the same time? This will prove the author's point more intuitively than theoretical arguments. If it is not an aligned evaluation criterion, can the author make the comparison as described above?

**Essential References Not Discussed:**

There are some missing references on both diffusion-model based adversarial robustness:

[1] Zhang J, Dong P, Chen Y, et al. Random Sampling for Diffusion-based Adversarial Purification[J]. arXiv preprint arXiv:2411.18956, 2024.

[2] Maiya S R, Ehrlich M, Agarwal V, et al. A frequency perspective of adversarial robustness[J]. arXiv preprint arXiv:2111.00861, 2021.

[3] Chen H, Dong Y, Wang Z, et al. Robust classification via a single diffusion model[J]. arXiv preprint arXiv:2305.15241, 2023.

[4] Mei H, Dong M, Xu C. Efficient Image-to-Image Diffusion Classifier for Adversarial Robustness[J]. arXiv preprint arXiv:2408.08502, 2024.

Among them, paper [1] is also an optimization algorithm for diffusion purification, and paper [2] also discusses the characteristics of adversarial attacks in the frequency domain. Papers [3] and [4] show another possibility of using diffusion models to improve adversarial robustness. Adding these works can more comprehensively demonstrate the research progress in this field.

**Experimental Designs Or Analyses:**

The experimental analysis of this paper is effective and reasonable

**Methods And Evaluation Criteria:**

The evaluation method used in this work is a commonly used evaluation strategy for purification and meets the experimental requirements.

**Other Comments Or Suggestions:**

No other comments

**Other Strengths And Weaknesses:**

The paper's discussion of the ideas, method and its structure are clear.
This method has a significant improvement in performance on various data sets.

**Questions For Authors:**

A more detailed discussion of the frequency domain characteristics of adversarial perturbations and a comparison with diffusion perturbations.

**Relation To Broader Scientific Literature:**

The authors' findings on the characteristics of adversarial attacks in the frequency domain may be generalizable and may guide a wide range of purification or AT learning processes.

**Theoretical Claims:**

1. Regarding the author's assertion that adversarial perturbations “increase monotonically” in the frequency domain, it is noted that the article [1] advances a perspective: “We demonstrate that adversarial examples are neither high frequency nor low frequency phenomena.” Does This present a contradiction to this work's view. How does the author reconcile these disparate views and phenomena?
[1] Maiya S R, Ehrlich M, Agarwal V, et al. A frequency perspective of adversarial robustness[J]. arXiv preprint arXiv:2111.00861, 2021.

2. Var in Equation 3 appears to be monotonically decreasing with respect to t, because it has a linear relationship with sqrt(alpha_t) and alpha_t, and alpha_t monotonically decreases with respect to t. This does not match the author's relationship, so please check and explain.

---

> ### Author Rebuttal · Authors · 2025-04-01
>
> Thank you for your constructive feedback, which will enhance the completeness and persuasiveness of our article.
> ## Claims And Evidence
> Figure 1 shows the extent of damage caused by adversarial perturbations to the phase spectrum and amplitude spectrum of images at different frequency components. The vertical axis represents the difference between the amplitude spectrum and phase spectrum of original clean samples and adversarial samples. It is not the Var mentioned in Section 3. To illustrate our theory in Section 3 more intuitively, we conduct experiments using Var as the vertical coordinate, and we provide the experimental results from low frequency to high frequency: (https://bashify.io/i/tBXTqh). These results are consistent with our theoretical analysis. Additionally, we find that the phase spectrum is more easily damaged compared to the amplitude spectrum, indicating the importance of preserving the phase spectrum during the purification process. Regarding final purification damage and attack damage of diffusion, we calculate the mean of different frequency variations for some samples. The experimental results are as follows: (https://bashify.io/i/CQpvZD). We observe that the trends of attack damage of diffusion and purification damage are consistent, which also supports our claims.
> ## Different experimental conclusions
> We cite this paper in the second paragraph of the Introduction and briefly claim the differences between these methods and ours. Here, we elaborate on the distinctions between this paper and our method. First, there is a difference in the measurement approach. This paper defines a Perturbation Gradients $\frac{dy}{d\delta}$ and observes its variation with frequency using the **DCT** decomposition. Therefore, more accurately, the conclusion of this paper should be that Perturbation Gradients are neither high frequency nor low frequency phenomena, while we directly measure the differences between the amplitude and phase spectrum values of adversarial samples in the frequency domain and those of normal samples. Additionally, the decomposition method we use is the **DFT**, which can directly compute the phase spectrum of images. The importance of the phase spectrum has been validated in many papers, such as  (Chen et al., CVPR2021) and (Zhou et al., ICML2023).
> ## Monotonicity of Equation 3
> The RHS being monotonically decreasing requires both coefficients to be greater than 0. However, under the assumptions we derived, the coefficient of the first term is less than 0. However, this assumption is somewhat strong, so we re-derive part of the conclusions in the proof. Specifically, we re-derive the first moment of the amplitude spectrum:
> $$
>     \mathbf{x}_t(u,v) = \sqrt{\overline{\alpha}_t}\mathbf{x}_0(u,v)+\sqrt{1-\overline{\alpha}_t}\mathbf{\epsilon}(u,v)=
> \underbrace{\mathfrak{R}\mathfrak{e}(\sqrt{\overline{\alpha}_t}\mathbf{x}_0(u,v))+\mathfrak{R}\mathfrak{e}(\sqrt{1-\overline{\alpha}_t}\mathbf{\epsilon}(u,v))}_R+i\underbrace{(\mathfrak{I}\mathfrak{m}(\sqrt{\overline{\alpha}_t}\mathbf{x}_0(u,v))+\mathfrak{I}\mathfrak{m}(\sqrt{1-\overline{\alpha}_t}\mathbf{\epsilon}(u,v)))}_I
> $$
> $R\sim\mathcal{N}(\mathfrak{R}\mathfrak{e}(\sqrt{\overline{\alpha}_t}\mathbf{x}_0(u,v)),\frac{1-\overline{\alpha}_t}{2})$ and $I\sim\mathcal{N}(\mathfrak{I}\mathfrak{m}(\sqrt{\overline{\alpha}_t}\mathbf{x}_0(u,v)),\frac{1-\overline{\alpha}_t}{2})$.
> We can see that the means of the real part and the imaginary part are different, the variances are the same, and they are independent of each other. Therefore, the amplitude $|\mathbf{x}_t(u,v)|$ follows a **Rice distribution**. "Therefore, we can utilize some known conclusions.With the assumption of $SNR_t>1$ in the Theorem 3.4. We can obtain the following conclusions：
> $$
>     \mathbb{E}(|\mathbf{x}_t(u,v)|)\approx \nu+\frac{\sigma^2}{2\nu}=\sqrt{\overline{\alpha}_t}|\mathbf{x}_0(u,v)| + \frac{1-\overline{\alpha}_t}{4\sqrt{\overline{\alpha}_t}|\mathbf{x}_0(u,v)|}
> $$
> For $\mathbb{E}(|\mathbf{x}_t(u,v)|^2)$ we still use the conclusion derived from Equation 40. We re-derive $Var(\Delta A_t(u,v))$as follows:
> $$
>     \begin{aligned}
>         Var(\Delta A_t(u,v))
>         &\approx \overline{\alpha}_t |\mathbf{x}_0(u,v)|^2+(1-\overline{\alpha}_t) + (\sqrt{\overline{\alpha}_t}|\mathbf{x}_0(u,v)| + \frac{1-\overline{\alpha}_t}{4\sqrt{\overline{\alpha}_t}|\mathbf{x}_0(u,v)|})^2\\
>         &=\frac{1-\overline{\alpha}_t}{2} -\frac{(1-\overline{\alpha}_t)^2}{16|\mathbf{x}_0(u,v)|^2\overline{\alpha}_t}
>     \end{aligned}
> $$
> It is clear that the RHS is monotonically increasing with respect to $t$.
> ## Discussion
> The characteristics of adversarial perturbations in the frequency domain are that the values of high-frequency components are greater than those of low-frequency components. Diffusion perturbations are normal Gaussian noise, and their amplitude and phase spectrum distributions in frequency domain conform to a normal distribution.
> ## Missing Reference
> We will add them in our paper.

---

### Official Review · Reviewer_22XE · 2025-03-14

**Overall Recommendation:** 3

**Summary:**

The paper focuses on adversarial defense methods, particularly addressing challenges in accurately and quickly calculating gradients, which is crucial for evaluating the effectiveness of defense mechanisms. The authors propose a method that significantly outperforms other approaches in terms of both standard and robust accuracy. The paper also explores the sensitivity of their defense method to the number of denoising steps in the surrogate process, providing experimental analysis to support their findings.

**Claims And Evidence:**

Yes, the claims made in the submission appear to be supported by clear and convincing evidence.

**Essential References Not Discussed:**

The references cited in the paper appear to be adequate for supporting the key contributions and findings.

**Experimental Designs Or Analyses:**

Yes

**Methods And Evaluation Criteria:**

Yes

**Other Comments Or Suggestions:**

See weaknesses.

**Other Strengths And Weaknesses:**

## Strenghts

1. This paper addresses a critical challenge in adversarial defense: improving both standard accuracy and robust accuracy against adversarial attacks. The experimental results show substantial improvements, such as a 15.04% increase in standard accuracy and a 41.01% increase in robust accuracy on the SVHN dataset, which is a notable advancement over existing methods.
2. The paper is well-structured and clearly presents its methodology, experiments, and results.
3. The paper provides extensive experimental validation across multiple datasets and attack scenarios, demonstrating the robustness and generalizability of the proposed method.

## Weaknesses

1. The paper demonstrates improvements in adversarial robustness, but there is limited discussion on computational efficiency. Since diffusion-based models are already computationally expensive, adding frequency-domain modifications might introduce further overhead.
2. Some theoretical claims, while backed by empirical observations, could benefit from more formal proofs (e.g., The assumption that low-frequency components are less affected by adversarial perturbations is based on empirical findings, but no rigorous theoretical proof is provided).

**Questions For Authors:**

1. Could the authors elaborate on the computational efficiency of the proposed method, particularly in terms of training and inference time compared to existing methods?
2. Can the authors provide a principled way to select optimal hyperparameters for different datasets or attack settings?
3. Does the method generalize well to real-world scenarios, such as adversarial examples crafted under distribution shifts or physical-world attacks (e.g., adversarial patches)?

**Relation To Broader Scientific Literature:**

The paper extends the scientific literature by combining insights from training-based and diffusion-based purification methods while introducing a frequency domain perspective to enhance robustness against adversarial attacks. This aligns with and advances prior findings in the field.

**Theoretical Claims:**

Yes

---

> ### Author Rebuttal · Authors · 2025-04-01
>
> Thank you for your valuable feedback, which will enhance the integrity of our paper. To address your concerns, we have provided additional theoretical proofs and experiments. We sincerely hope our response resolves the concerns raised, and we would greatly appreciate reconsideration of the score.
> ## W1&Q1 Computational Efficiency
> Adversarial purification is a test-time defense that involves no training process, only an inference process. For example, for the ImageNet and CIFAR-10 datasets, we directly use the pre-trained weights of the corresponding diffusion models. In terms of inference time, our approach requires just one additional DFT/IDFT pair per time step, with minimal computational impact as the Fourier transform operations contribute negligible overhead. The inference time for various methods are as follows:
> | Method| Lee&Kim,2023| Baietal.,2024|Nieetal.,2022|Ours|
> | :---: |  :---:      | :---:       |:---:         |:---:|
> | Time(s)  | 19.09±0.04  |6.43±0.13    |4.26±0.14     |4.29±0.09|
> ## W2 Theoretical Proof
> We provide a simple proof regarding how adversarial perturbations primarily disrupt the high-frequency components of an image.
> The original image is denoted as $x$,  the adversarial perturbation as $\delta$ and $F$ represents the Fast Fourier Transform.
> The power spectrum characteristics of natural images follow the distribution as follows, where $\alpha>1$:
> $$
>     |F_x(w)|^2\propto \frac{1}{w^{\alpha}}\\
> $$
> The definition of the noise to signal ratio (NSR) is as follows:
> $$
>     NSR(w)=\frac{|F_{\delta(w)}|^2}{|F_x(w)|^2}
> $$
> Combining the above two formulas leads to the following relation:
> $$
>     NSR(w)\propto w^{\alpha}|F_{\delta}(w)|^2
> $$
> We choose $\alpha=2$ , and we assume that the attack objective is to maximize the NSR:
> $$
> \max\sum_{w}NSR(w)=\max\sum_{w} w^{2}|F_{\delta}(w)|^2
> $$
>  The above optimization objective should be satisfied when it is maximized:
> $$
>     |F_{\delta}(w)|^2\propto w^2
> $$
> We find that the power of the perturbation in the frequency domain is proportional to the square of the frequency, meaning that the perturbation tends to disrupt the high-frequency information of the image.
> ## Q2  A principled way to select optimal hyperparameters
> We first observe Figure 1 to determine the approximate range where adversarial perturbations cause minimal damage. Within this range, for smaller datasets, we tend to use grid search to find the optimal hyperparameters. For larger datasets, such as ImageNet, we proportionally expand the optimal hyperparameters found on the smaller datasets to search for the optimal hyperparameters.
> Specifically, directly multiply by 2 or 3 at the same time and then perform a grid search around it.
> ## Q3 Generalization to real-world scenarios
> To demonstrate the generalizability of our method, we conduct relevant experiments in the context of adversarial patch. The method for constructing adversarial patch is described in [1]. The dataset is ImageNet, and the classifier is ResNet50. Since different attack methods do not affect standard accuracy, we only test robust accuracy. The results are as follows:
> | Method| Lee&Kim,2023| Baietal.,2024|Nieetal.,2022|Ours|
> | :---:          |  :---:      | :---:       |:---:         |:---:|
> | Robust Acc(%)  |74.219 |82.812    |78.906              |83.594|
>
> [1] Brown T B, Mané D, Roy A, et al. Adversarial patch[J]. arXiv preprint arXiv:1712.09665, 2017.

---

> > ### Comment · Reviewer_22XE · 2025-04-03
> >
> > The authors address my concerns. I am raising my rating to 3.

---

> > > ### Author Response · Authors · 2025-04-03
> > >
> > > We are glad that our response address your concerns. Thank you for your review and recognition.

---

### Official Review · Reviewer_PfQW · 2025-03-19

**Overall Recommendation:** 3

**Summary:**

The paper proposes a novel adversarial purification method called FreqPure through frequency domain analysis and theoretical proof. The core idea of the method is to provide effective prior guidance for image purification by selectively retaining low-frequency spectral information. Experimental results demonstrate its significant advantages in eliminating adversarial perturbations while preserving semantic information.

**Claims And Evidence:**

The article conducts comparative experiments with other methods, providing quantitative experimental data and visualization results.

**Essential References Not Discussed:**

The key contribution of the paper is proposing a purification method that can eliminate adversarial perturbations while maximizing the preservation of the content and structure of the original image. The approach from the frequency domain is a novel perspective, and the authors provide mathematical proofs for its rationality and effectiveness.

**Experimental Designs Or Analyses:**

I reviewed the main experimental results, ablation experiments, and supplementary experiments in the appendix. The experimental setup follows the settings of previous work. However, in the ablation experiments, I am puzzled as to why the results of ASE+PSE without PSP are not included.

**Methods And Evaluation Criteria:**

The proposed methods and evaluation criteria can effectively address the current problem and provide a reasonable basis for measuring related applications.

**Other Comments Or Suggestions:**

In line 256, you intended to reference ​Algorithm 1 in Section 4.3, but the PDF displays ​4.3. Please verify whether the reference number is correct.

**Other Strengths And Weaknesses:**

Strengths:
(1) This paper analyzes the gap between adversarial images and original images from the frequency domain perspective, and provides derivations and proofs of the related formulas.
(2) In the experiments, the proposed method shows significant improvements over the baselines, demonstrating its effectiveness.
(3) Extensive visualization results effectively confirm that the purified images are closer to the original images.
Weaknesses:
Some details in the experimental setup and ablation studies are not clearly explained.

**Questions For Authors:**

(1) In the ablation study, the results for ASE+PSE without PSP are not included.
To strengthen the validity of the conclusions, the ablation study should include a comparison of ASE+PSE without PSP.
(2) Regarding the experimental setup, in Algorithm 1, t = t*, …, 1 is mentioned, while in the experiments, values like t* = 0.2, t* = 0.3, and t* = 0.4 are used. This creates confusion for readers about the specific meaning of t*.
Additionally, the evaluation metrics, Standard Acc and Robust Acc, are not clearly defined. While it is mentioned that "standard accuracy is calculated on clean images, and robust accuracy is assessed on adversarial examples," a more detailed explanation of how these metrics are computed and their significance would improve clarity.
(3) In the experimental tables, one baseline method uses only half the number of iterations due to computational overhead. However, the number of iterations is a critical factor for PGD, and reducing it by half may weaken the strength of the PGD attack, leading to an unfair comparison of Robust Acc.
To address this, either a reasonable explanation should be provided for the reduced iterations, or experimental data with the same number of iterations should be included to ensure a fair comparison.
(4) While the method’s effectiveness is demonstrated in terms of Standard and Robust Acc, it is also important to evaluate the efficiency of the proposed method.
A comparison with other methods in terms of memory and computational overhead would be a valuable addition to the experimental section.

**Relation To Broader Scientific Literature:**

The method proposed in this paper decomposes images into amplitude and phase spectra and explores how to perform image restoration on adversarial images. It represents another approach to improving model robustness besides adversarial training.

**Theoretical Claims:**

The overall proof logic of the paper is rigorous, and the derivation process is reasonable. Specifically, for Theorem 3.2, "the variance of the difference of amplitude at time-step t between the clean image x0​ and the noisy image xt​" is provided with a complete derivation process in the appendix. This indicates that the authors have provided a detailed and rigorous explanation of the proof for Theorem 3.2, ensuring its correctness and reliability.

---

> ### Author Rebuttal · Authors · 2025-03-29
>
> Thank you for your careful review and valuable feedback. We have made every effort to address your concerns. We believe that investigating diffusion model-based adversarial purification from a frequency-domain perspective enables further research.
> ## Q1 Results for ASE+PSE without PSP
> Thank you for your careful review. We indeed overlook this situation, and we have added the relevant experiments.  The complete ablation experiment table is as follows:
> |ASE    |PSP    |PSE   |Standard| Robust|
> | :---: | :---: | :---:|:---:   |:---:|
> |x      |x      |x     | 87.89  |53.52|
> |✓      |x      |x     |90.82  |87.11|
> |x      |x     |✓      | 94.14 |79.30|
> |x      |✓     | x     | 94.53 |80.47|
> |✓      |x     |✓      |93.36|87.50|
> |✓      |✓    |x       | 94.53 |88.28|
>
> The last line, ASE+PSE, represents our complete method.
> ## Q2 Meaning of t*
> In our paper, $t^*$ does not refer to the optimal value but rather to a hyperparameter. The adversarial purification based on diffusion models can be roughly divided into two stages. The first stage is the forward process, where noise is added; the role of $t^*$  is to control the intensity of the noise added. The second stage is the denoising process. Here, we select three different values for DiffPure on the ImageNet dataset to explore the best performance of the method under different hyperparameters for comparison.
> ## Q2 A detailed explanation of evaluation metrics
> The purpose of adversarial purification is to remove the adversarial perturbations from adversarial samples so that these samples can be classified correctly as much as possible, while minimizing the impact on the classification of normal samples.
> Let normal samples be represented by $x$ and adversarial samples by $x_{\text{adv}}$. The purification method is denoted as $\text{AP}$, and the classifier is represented by $f$.
> For **Standard Accuracy**, we select a batch of normal samples and calculate whether the predicted labels $f(x)$ and $f(\text{AP}(x))$ are consistent. The Standard Accuracy is then computed as the number of consistent predictions divided by the total number of normal samples.
> For **Robust Accuracy**, we select a batch of adversarial samples and calculate whether the predicted labels $f(x_{\text{adv}})$ and $f(\text{AP}(x_{\text{adv}}))$ are consistent. The Robust Accuracy is computed as the number of consistent predictions divided by the total number of adversarial samples.
> The number of samples we select is consistent with previous methods, and we conduct multiple rounds of experiments while calculating the errors.
> ## Q3 Number of iterations about Baietal.,2024
> The more iterations there are, the stronger the attack effect becomes, which leads to a decrease in robust accuracy. For the method proposed by Bai et al. (2024), the robust accuracy with half the number of iterations is lower than that of other methods. Therefore, the robust accuracy with the complete number of iterations will be even lower. In Table 1, the robust accuracy decreases from 49.22% to 48.92% under the full number of iterations.
> ## Q4 Memory and computational overhead
> Compared to other methods, our approach introduces an additional discrete Fourier transform and inverse discrete Fourier transform only once at each time step, and the time introduced by the Fourier transform is almost negligible. The inference times and memory used for various methods are as follows:
> | Method| Lee&Kim,2023| Baietal.,2024|Nieetal.,2022|Ours|
> | :---: |  :---:      | :---:       |:---:         |:---: |
> | Time(s)  | 19.09±0.04  |6.43±0.13    |4.26±0.14     |4.29±0.09|
> | Memory(GB)|        0.56| 0.70        | 0.56      |   0.56|
> ## Suggestions: false reference number
> Thank you for noticing this issue. We have made corrections and will check and correct other typos.

---

### Decision · Program_Chairs · 2025-05-01

**Decision:**

Accept (spotlight poster)

**Comment:**

Based on the observations that the diffusion-based adversarial purification methods attempt to drown adversarial perturbations into a part of isotropic noise through the forward process, and then recover the clean images through the reverse process, and that due to the lack of distribution information about adversarial perturbations in the pixel domain, it is often unavoidable to damage normal semantics, the authors in this paper turn to the frequency domain perspective, decomposing the image into amplitude spectrum and phase spectrum, and find that for both spectra, the damage caused by adversarial perturbations tends to increase monotonically with frequency, meaning that it is possible to extract the content and structural information of the original clean sample from the frequency components that are less damaged. Besides, the theoretical analysis indicates that existing purification methods indiscriminately damage all frequency components, leading to excessive damage to the image.
Based on the above concerns, the authors propose a purification method that can eliminate adversarial perturbations while maximizing the preservation of the content and structure of the original image.

During the review and rebuttal processes, all the four reviewers consistently give positive comments on this work, and the raised questions have been properly addressed.
Therefore, this paper is suggested to be accepted in ICML'25.